# Lipoprotein DolP supports proper folding of BamA in the bacterial outer membrane promoting fitness upon envelope stress

David Ranava[1†], Yiying Yang[1†], Luis Orenday-Tapia[1†], François Rousset[2], Catherine Turlan[1], Violette Morales[1], Lun Cui[2], Cyril Moulin[1], Carine Froment[3], Gladys Munoz[1], Jérôme Rech[1], Julien Marcoux[3], Anne Caumont-Sarcos[1], Cécile Albenne[1], David Bikard[2], Raffaele Ieva[1*]

[1]Laboratoire de Microbiologie et Génétique Moléculaires (LMGM), Centre de Biologie Intégrative (CBI), Université de Toulouse, CNRS, UPS, Toulouse, France; [2]Synthetic Biology Group, Microbiology Department, Institut Pasteur, Paris, France; [3]Institut de Pharmacologie et de Biologie Structurale (IPBS), Université de Toulouse, CNRS, UPS, Toulouse, France

**Abstract** In Proteobacteria, integral outer membrane proteins (OMPs) are crucial for the maintenance of the envelope permeability barrier to some antibiotics and detergents. In Enterobacteria, envelope stress caused by unfolded OMPs activates the sigmaE ($\sigma^E$) transcriptional response. $\sigma^E$ upregulates OMP biogenesis factors, including the β-barrel assembly machinery (BAM) that catalyses OMP folding. Here we report that DolP (formerly YraP), a $\sigma^E$-upregulated and poorly understood outer membrane lipoprotein, is crucial for fitness in cells that undergo envelope stress. We demonstrate that DolP interacts with the BAM complex by associating with outer membrane-assembled BamA. We provide evidence that DolP is important for proper folding of BamA that overaccumulates in the outer membrane, thus supporting OMP biogenesis and envelope integrity. Notably, mid-cell recruitment of DolP had been linked to regulation of septal peptidoglycan remodelling by an unknown mechanism. We now reveal that, during envelope stress, DolP loses its association with the mid-cell, thereby suggesting a mechanistic link between envelope stress caused by impaired OMP biogenesis and the regulation of a late step of cell division.

**\*For correspondence:**
raffaele.ieva@univ-tlse3.fr

[†]These authors contributed equally to this work

**Competing interests:** The authors declare that no competing interests exist.

## Introduction

The outer membrane (OM) of Gram-negative bacteria forms a protective barrier against harmful compounds, including several antimicrobials. This envelope structure surrounds the inner membrane and the periplasm that contains the peptidoglycan, a net-like structure made of glycan chains and interconnecting peptides. During cell division, the multi-layered envelope structure is remodelled by the divisome machinery (*den Blaauwen et al., 2017*). At a late step of division, septal peptidoglycan synthesized by the divisome undergoes splitting, initiating the formation of the new poles of adjacent daughter cells. Finally, remodelling of the OM barrier completes formation of the new poles in the cell offspring. The mechanisms by which cells coordinate OM remodelling with peptidoglycan splitting, preserving the permeability barrier of this protective membrane, are ill-defined (*Egan et al., 2020*).

Integral outer membrane proteins (OMPs) are crucial to maintain the OM permeability barrier. OMPs fold into amphipathic β-barrel structures that span the OM and carry out a variety of tasks. Porins are OMPs that facilitate the diffusion of small metabolites. Other OMPs function as cofactor transporters, secretory channels, or machineries for the assembly of proteins and lipopolysaccharide

(LPS), a structural component of the external OM leaflet that prevents the diffusion of noxious chemicals (*Calmettes et al., 2015*; *Nikaido, 2003*). The β-barrel assembly machinery (BAM) is a multi-subunit complex that mediates the folding and membrane insertion of OMPs transiting through the periplasm (*Ranava et al., 2018*; *Schiffrin et al., 2017*). The essential and evolutionarily conserved BamA insertase subunit is an OMP consisting of an amino (N)-terminal periplasmic domain made of polypeptide transport-associated (POTRA or P) motifs and a carboxy (C)-terminal 16-stranded β-barrel membrane domain that catalyses OMP biogenesis (*Ranava et al., 2018*). The flexible pairing of β-strands 1 and 16 of the BamA β-barrel controls a lateral gate connecting the interior of the barrel towards the surrounding lipid bilayer (*Bakelar et al., 2016*; *Gu et al., 2016*; *Iadanza et al., 2016*; *Noinaj et al., 2013*). Conformational dynamics of the BamA β-barrel region proximal to the lateral gate is thought to locally increase the entropy of the surrounding lipid bilayer (*Doerner and Sousa, 2017*; *Noinaj et al., 2013*) and to assist the insertion of nascent OMPs into the OM (*Doyle and Bernstein, 2019*; *Gu et al., 2016*; *Tomasek et al., 2020*). The N-terminal periplasmic portion of BamA from the enterobacterium *Escherichia coli* contains five POTRA motifs that serve as a scaffold for four lipoproteins, BamBCDE, which assist BamA during OMP biogenesis (*Kim et al., 2007*; *Sklar et al., 2007*; *Wu et al., 2005*). The N-terminal POTRA motif is also the docking site for the periplasmic chaperone SurA (*Bennion et al., 2010*). Together with the chaperones Skp and DegP, SurA contributes to monitor unfolded OMPs transported into the periplasm by the inner membrane general secretory (Sec) apparatus (*Crane and Randall, 2017*; *Rizzitello et al., 2001*).

Defective OMP assembly causes periplasmic accumulation of unfolded protein transport intermediates. This envelope stress is signalled across the inner membrane to induce the sigmaE ($\sigma^E$)-mediated transcriptional response (*Walsh et al., 2003*). In the absence of a stress, $\sigma^E$ is sequestered by the inner membrane-spanning RseA factor. By-products of misfolded OMP turnover activate degradation of RseA, liberating $\sigma^E$ (*Ades, 2008*). The $\sigma^E$ response copes with stress (i) by upregulating genes involved in OMP biogenesis, such as the *bam* genes (*Rhodius et al., 2006*), and (ii) by lowering the OMP biogenesis burden via a post-transcriptional mechanism (*Guillier et al., 2006*). Whereas $\sigma^E$ is essential (*De Las Peñas et al., 1997a*), a tight control of cytosolic $\sigma^E$ availability is necessary for optimal cell fitness and to prevent a potentially detrimental effect on the envelope (*De Las Peñas et al., 1997b*; *Missiakas et al., 1997*; *Nicoloff et al., 2017*). Remarkably, the functions of a number of genes upregulated by $\sigma^E$ remain unknown. Among those, *dolP/yraP* (recently renamed division and OM stress-associated lipid-binding protein) encodes an ~20 kDa OM-anchored lipoprotein largely conserved in γ and β proteobacteria that is crucial for OM integrity and pathogenicity (*Bos et al., 2014*; *Bryant et al., 2020*; *Morris et al., 2018*; *Onufryk et al., 2005*; *Seib et al., 2019*). DolP consists of two consecutive BON (bacterial OsmY and nodulation) domains, a family of conserved folding motifs named after the osmotic stress-induced periplasmic protein OsmY (*Yeats and Bateman, 2003*). During a late step of cell division, DolP localizes at the mid-cell where it contributes to the regulation of septal peptidoglycan splitting by an unknown mechanism (*Tsang et al., 2017*). A recent structural analysis of DolP reveals a phospholipid-binding site in the C-terminal BON domain (*Bryant et al., 2020*). It remains unclear, however, why DolP is upregulated in response to $\sigma^E$ activation and how this lipoprotein helps coping with envelope stress.

By using a genome-wide synthetic-defect screen, we show that DolP is particularly important when the BAM complex is defective and under envelope stress conditions. We demonstrate that DolP interacts with the BAM complex in the OM and supports the proper folding and functioning of the BamA subunit. Taken together our results indicate that DolP functions as a fitness factor during activation of the $\sigma^E$ response and that BamA is a molecular target of the fitness role of DolP. We also reveal that, upon envelope stress, DolP loses its association with the mid-cell, thus suggesting a possible link between the envelope stress response and septal peptidoglycan hydrolysis during a late step of cell division.

## Results

### A genome-wide synthetic-defect screen identifies *dolP* genetic interactions

The mutant allele Δ*dolP::kan* (*Baba et al., 2006*) was introduced into *E. coli* BW25113 by P1 transduction. The resulting Δ*dolP* strain grew normally on LB medium, but was highly susceptible to

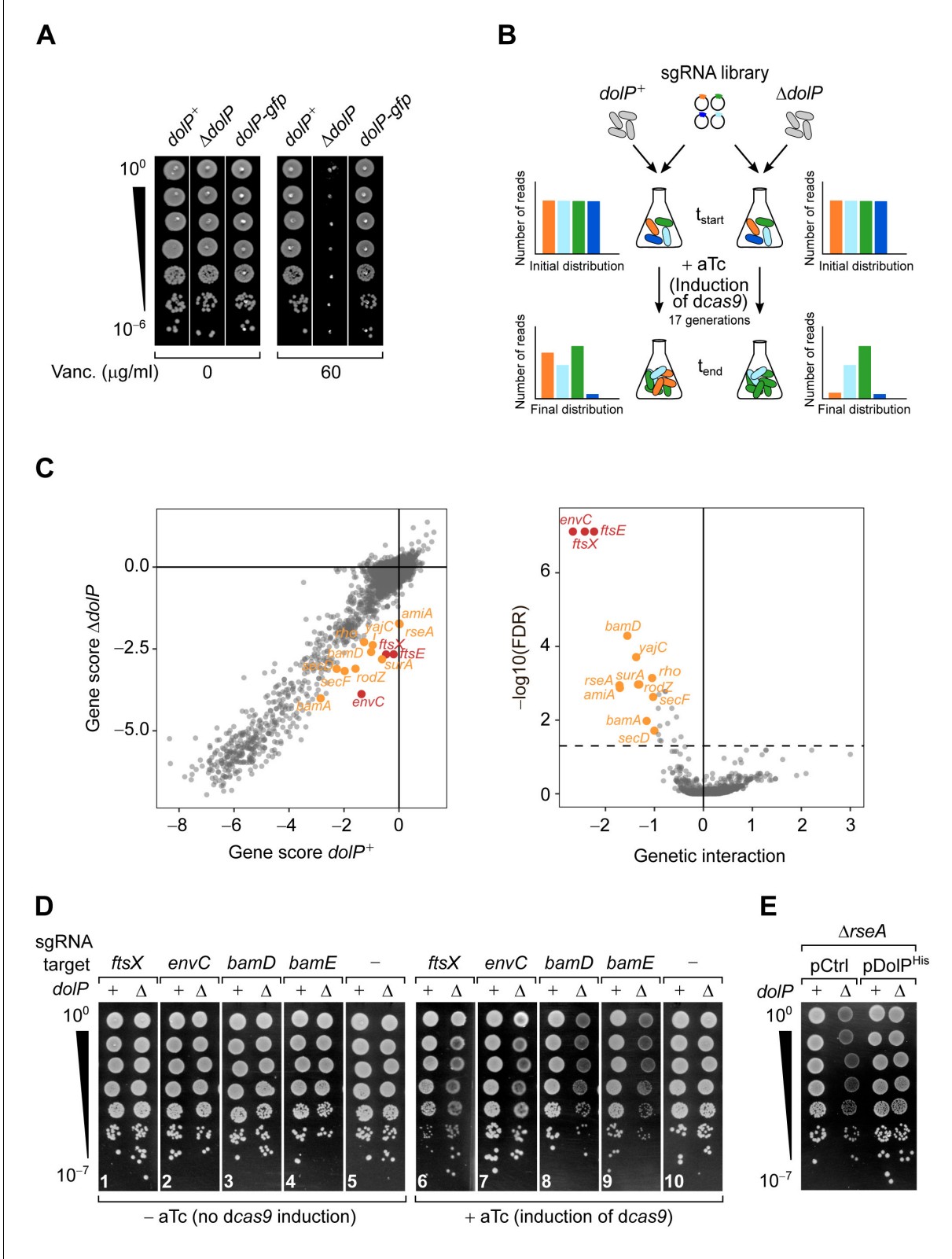

**Figure 1.** Genome-wide screen of *dolP* genetic interactions. (**A**) The deletion of *dolP* impairs OM integrity. The indicated strains were serially diluted and spotted onto LB agar plates lacking or supplemented with 60 µg/ml vancomycin as indicated. (**B**) Schematic representation of the CRISPR-based gene silencing approach. LC-E75 (*dolP*⁺) or its Δ*dolP* derivative strain, both carrying d*cas9* under the control of an anhydrotetracycline (aTc)-inducible promoter in their chromosome were transformed with a library of plasmids encoding gene-specific sgRNAs. The library covers any *E. coli* MG1655

*Figure 1 continued on next page*

*Figure 1 continued*

genetic features with an average of five sgRNAs per gene. Pooled transformed cells were cultured to early exponential phase prior to plasmid extraction and quantitative Illumina sequencing to assess the initial distribution of sgRNA constructs in each culture ($t_{start}$). Upon addition of 1 µM aTc to induce sgRNA-mediated targeting of dcas9 for approximately 17 generations, samples of cells from each culture were newly subjected to plasmid extraction and Illumina sequencing to determine the final distribution of sgRNA constructs ($t_{end}$). (**C**) Left: Comparison of gene scores obtained in *dolP*⁺ and Δ*dolP* screens. The log2 fold-change (log2FC) between $t_{end}$ and $t_{start}$ calculated for each sgRNAs (*Figure 1—figure supplement 2B*) was grouped by gene target, and their median was used to derive fitness gene scores (see also *Figure 1—source data 1* and *2*). Right: Volcano plot of the *dolP* genetic interaction scores. The x-axis shows a genetic interaction score calculated for each gene based on the minimum hypergeometric (mHG) test conducted on the ranked difference of sgRNA-specific log2FC values between the Δ*dolP* and the *dolP*⁺ screens. The y-axis shows the log10 of the false discovery rate (FDR) of the test. The dashed line shows FDR = 0.05. In both panels, genes highlighted in orange have FDR < 0.05 and GI >1 whereas genes highlighted in red have FDR < 0.05 and GI > 2. (**D and E**) Validation of the genetic interactions determined in (**C**). (**D**) LC-E75 (*dolP*⁺) or its Δ*dolP* derivative strain expressing sgRNAs that target the indicated genes were serially diluted and spotted on LB agar lacking or supplemented with aTc to induce expression of dcas9, as indicated. (**E**) BW25113 derivative cells deleted of *rseA* or both *rseA* and *dolP* were transformed with an empty vector (pCtrl) or a plasmid encoding DolP (pDolPᴴⁱˢ). Ectopic expression of DolPᴴⁱˢ was driven by the leaky transcriptional activity of P$_{trc}$ in the absence of IPTG. (**D and E**) Ten-fold serial dilutions of the indicated transformants were spotted on LB agar.

The online version of this article includes the following source data and figure supplement(s) for figure 1:

**Source data 1.** Log2FC values of sgRNAs in the screens conducted with wild-type or Δ*dolP* cells.
**Source data 2.** Genetic interaction scores.
**Figure supplement 1.** The deletion of *dolP* severely impairs growth in the presence of vancomycin.
**Figure supplement 2.** Reproducibility of the CRISPRi screens and ontology analysis of gene hits.

vancomycin (*Figure 1A* and *Figure 1—figure supplement 1A*). This antibiotic is normally excluded from the OM of wild-type cells but inhibits growth of cells lacking OMP biogenesis factors such as *skp* and *surA* (*Figure 1—figure supplement 1B*). The expression of C-terminally tagged DolP protein variants in place of its wild-type form restored vancomycin resistance (*Figure 1A* and *Figure 1—figure supplement 1C*). This result supports the notion that DolP is important for envelope integrity (*Bos et al., 2014*; *Onufryk et al., 2005*; *Seib et al., 2019*; *Tsang et al., 2017*). However, the role of DolP during envelope stress remains poorly understood.

To gain insights into the role of DolP, we subjected Δ*dolP* cells to a genome-wide synthetic-defect screen exploiting a Clustered Regularly Interspaced Short Palindromic Repeat interference (CRISPRi) approach. Targeting of the catalytically inactive dCas9 nuclease by gene-specific single guide RNAs (sgRNAs) enables gene repression (*Figure 1B*; *Cui et al., 2018*). The EcoWG1 sgRNA library targeting the entire genome of *E. coli* MG1655 (*Calvo-Villamañán et al., 2020*) was introduced into isogenic Δ*dolP* or *dolP*⁺ MG1655-derivative strains. The fitness of each knockdown was then compared in these backgrounds by deep-sequencing of the sgRNA library after ~17 growth generations. The outputs obtained from two independent tests were highly reproducible (*Figure 1—figure supplement 2A*). A strong fitness defect in the Δ*dolP* strain was caused by the targeting of *envC*, followed by the targeting of *ftsX* and *ftsE* (*Figure 1C*, *Figure 1—figure supplement 2B*, *Figure 1—source data 1* and *2*). A validation growth test showed that the synthetic fitness defect observed for Δ*dolP* cells was caused by dCas9-dependent silencing of *ftsX* and *envC* (*Figure 1D*, panels 6 and 7). The ABC transporter-like complex FtsE/FtsX has multiple roles in organizing the cell divisome, including the recruitment of periplasmic EnvC, a LytM domain-containing factor required for the activation of amidases that hydrolyse septal peptidoglycan (*Pichoff et al., 2019*). This peptidoglycan remodelling reaction is mediated by two sets of highly controlled and partially redundant amidases, AmiA/AmiB and AmiC (*Heidrich et al., 2001*; *Uehara et al., 2009*). Whereas AmiA and AmiB are activated at the inner membrane/peptidoglycan interface by the coordinated action of FtsE/FtsX and EnvC, activation of AmiC requires the OM-anchored LytM domain-containing lipoprotein NlpD (*Uehara et al., 2010*; *Yang et al., 2011*). Under laboratory conditions, the activity of only one of these two pathways is sufficient for septal peptidoglycan splitting, whereas inhibition of both pathways leads to the formation of chains of partially divided cells, i.e., cells that have begun to divide but that are blocked at the step of septal peptidoglycan splitting (*Uehara et al., 2010*). A recent report showed that *dolP* is necessary for completion of septal peptidoglycan splitting and cell separation when the AmiA/AmiB pathway is inactive, somehow linking DolP to AmiC activation (*Tsang et al., 2017*). Thus, the reduced fitness caused by silencing of *envC*, *ftsE*, or *ftsX* in Δ*dolP* cells (*Figure 1C*) can be explained by the impaired cell separation when both the AmiA/AmiB and

the AmiC pathways are not active. In keeping with this notion, *amiA* itself was found among the negative fitness hits of the CRISPRi screen (*Figure 1C*). The *amiB* gene was not a hit (*Figure 1—source data 2*) probably because AmiA is sufficient to split septal peptidoglycan in the absence of other amidases (*Chung et al., 2009*).

Most importantly, the CRISPRi approach identified novel *dolP*-genetic interactions that had a score similar to that obtained for *amiA* (*Figure 1C*). These included an interaction with *rseA*, encoding the inner membrane σ$^E$-sequestering factor, as well as with *bamD*, encoding an essential subunit of the BAM complex (*Malinverni et al., 2006*; *Onufryk et al., 2005*). In accordance with the screen output, a serial dilution assay confirmed that CRISPRi reducing the levels of BamD or of its stoichiometric interactor BamE (*Figure 1—figure supplement 2D*) causes a fitness defect in cells lacking DolP (*Figure 1D*, panels 8 and 9). In addition, the interaction of *dolP* with *rseA* was confirmed in the genetic background of a BW25113 strain (*Figure 1E*). Further genes, involved in OMP biogenesis and more generally in protein secretion, had a lower interaction score (*Figure 1—source data 2* and *Figure 1—figure supplement 2C*) and are highlighted in *Figure 1C*. These comprise *bamA*, the OMP chaperone-encoding gene *surA*, as well as the genes encoding the Sec ancillary complex SecDF-YajC that contributes to efficient secretion of proteins including OMPs (*Crane and Randall, 2017*) and that was shown to interact with the BAM complex (*Alvira et al., 2020*; *Carlson et al., 2019*). Collectively, the results of the CRISPRi screen indicate that the function of DolP is particularly critical for cell fitness upon inactivation of septal peptidoglycan hydrolysis by AmiA, as well as when the BAM complex is defective or the assembly of proteins in the OM is impaired.

## DolP improves cell fitness when the OM undergoes stress

The newly identified genetic interaction between *dolP* and *bamD* (*Figure 1C and D*) points to a possible role of DolP in OM biogenesis. However, the overall protein profile of the crude envelope fraction was not affected by the deletion of *dolP* (*Figure 2—figure supplement 1A*, lanes 1–4). OMPs such as the abundant OmpA and OmpC (*Li et al., 2014*) can be recognized by the characteristic heat-modifiable migration patterns of their β-barrel domains when separated by SDS-PAGE (*Figure 2—figure supplement 1A–C*; *Nakamura and Mizushima, 1976*). The envelope protein profiles were not affected also when *dolP* was deleted in cells lacking one of the OMP periplasmic chaperones Skp or DegP (*Figure 2—figure supplement 1A*, lanes 5–12). Furthermore, the levels of BamA and BamE, which are susceptible of σ$^E$-mediated regulation, were not increased in Δ*dolP* cells (*Figure 2A*). Taken together, these observations suggest that if DolP plays a role in OM biogenesis, this would probably be indirectly related to the process of OMP assembly.

To further test the role of DolP under envelope stress conditions, we deleted *dolP* in a strain lacking *bamB*, which was identified with a lower genetic score by the CRISPRi approach (*Figure 1—source data 2*). In Δ*bamB* cells, the σ$^E$ response is partially activated (*Charlson et al., 2006*; *Wu et al., 2005*), causing the upregulation of *bam* genes (*Figure 2B*). A strain carrying the simultaneous deletion of *dolP* and *bamB* was viable but growth-defective. Normal growth was restored by ectopic expression of a C-terminally polyhistidine tagged DolP protein variant (*Figure 2C*). As expected, the Δ*bamB* envelope protein profile presented a marked reduction of the major heat-modifiable OMPs, OmpA and OmpC. The concomitant lack of DolP had no additional effect on the levels of these OMPs (*Figure 2—figure supplement 1A*, lanes 13–18). However, we noticed that the levels of BamA, which presents the typical heat modifiable behaviour of OMPs, were reduced in this strain (*Figure 2D*). Furthermore, phase-contrast microscopy analysis of the same Δ*dolP* Δ*bamB* strain revealed a number of cells with altered morphology (*Figure 2E*). Taken together, these results corroborate the importance of DolP when the BAM complex is defective and point to a possible role of DolP in maintaining BamA levels during envelope stress.

## DolP supports proper folding and function of BamA

In part because we found DolP to be critical in cells with an impaired BAM complex and in part because the BAM complex is upregulated upon activation of the σ$^E$ response, we wished to explore the effect of BAM overproduction in *dolP*$^+$ and Δ*dolP* cells. To this end, the genes encoding wild-type BamABCD and a C-terminally polyhistidine-tagged BamE protein variant were ectopically expressed via the isopropylthiogalactoside (IPTG)-inducible *trc* promoter (P$_{trc}$) as a transcriptional unit, adapting a previously established method (*Roman-Hernandez et al., 2014*). With 400 μM

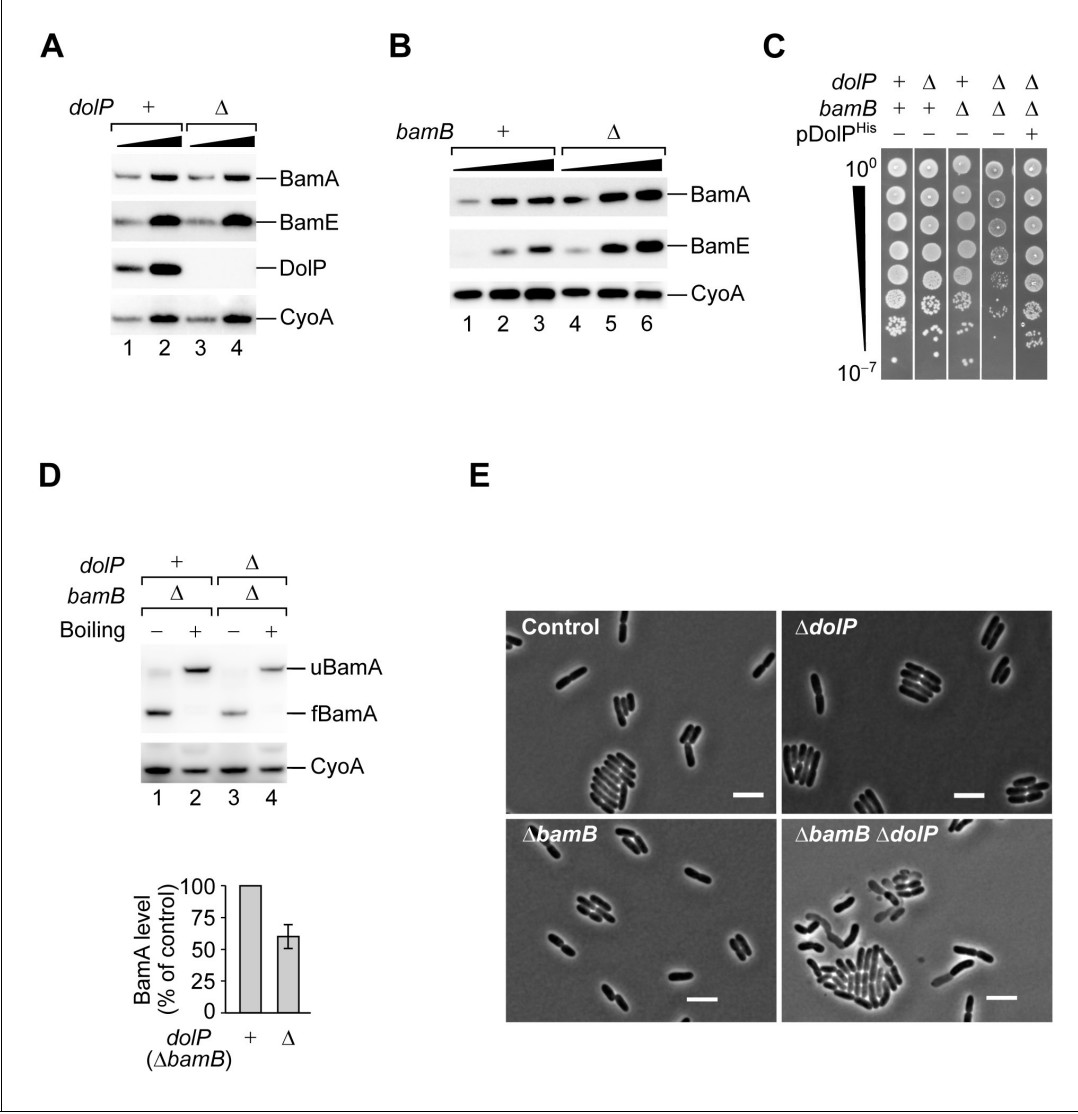

**Figure 2.** DolP promotes fitness in cells that undergo envelope stress. (**A**) One- and three-fold amounts of the total cell lysate fractions obtained from a BW25113 (*dolP*+) strain and a derivative Δ*dolP* strain were analysed by SDS-PAGE and immunoblotting using the indicated antisera. (**B**) One-, two-, and three-fold amounts of the total cell lysate fractions obtained from a BW25113 (*bamB*+) strain and a derivative Δ*bamB* strain were analysed by SDS-PAGE and immunoblotting using the indicated antisera. (**C**) BW25113 and derivative cells deleted of *dolP, bamB,* or both genes were cultured, serially diluted, and spotted on LB agar. Cells deleted of both *dolP* and *bamB* and transformed with pDolP^His were cultured, serially diluted, and spotted on LB agar supplemented with ampicillin. (**D**) The envelope fractions of the indicated strains were analysed by SDS-PAGE and immunoblotting. Prior to gel loading, samples were incubated at 25°C (Boiling −) or 99°C (Boiling +). The total amounts of BamA in Δ*bamB dolP*+ and Δ*bamB* Δ*dolP* strains were quantified, normalized to the amount of the inner membrane protein CyoA, and expressed as percentage of the value obtained for the Δ*bamB dolP*+ sample. Data are reported as means ± standard error of the mean (SEM, N = 3). u, unfolded; f, folded. (**E**) Overnight cultures of BW25113 (control), Δ*dolP,* Δ*bamB,* and Δ*dolP* Δ*bamB,* were freshly diluted in LB medium and re-incubated at 30°C until OD$_{600}$ = 0.3. Cells were visualized on 1% (w/v) agarose pads by phase contrast microscopy. Bar = 5 μm.

The online version of this article includes the following figure supplement(s) for figure 2:

**Figure supplement 1.** DolP does not play a direct role in OMP biogenesis.

IPTG, the amounts of BAM subunits that accumulated in the cell membrane fraction were roughly similar to those of the major OMPs OmpA or OmpC (*Figure 3A*, lane 2). Importantly, we noticed that BAM overproduction caused a partial detrimental effect in the wild-type BW25113 strain (*Figure 3B*). The detrimental effect was more severe in a Δ*dolP* strain (*Figure 3B*). This difference was particularly noticeable with 200 μM IPTG, which had a minor inhibitory effect on the growth of wild-type cells but strongly impaired the growth of a Δ*dolP* strain. Similar to *dolP*, *skp* is upregulated

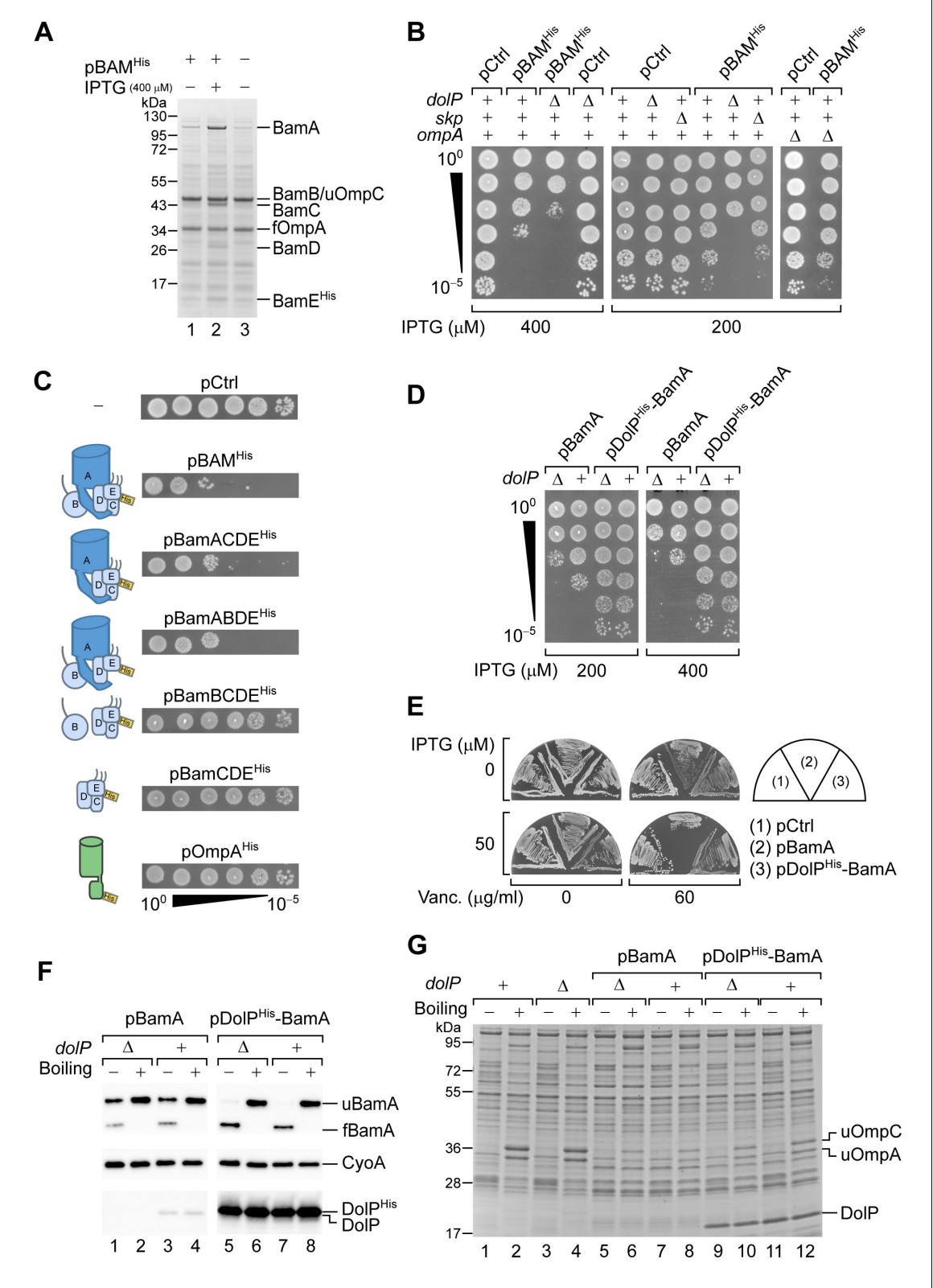

**Figure 3.** DolP opposes an envelope detrimental effect caused by BamA overaccumulation in the OM. (**A**) BW25113 cells harbouring pBAM^His where indicated were cultured and supplemented with no IPTG or 400 μM IPTG for 1 hr prior to collecting cells. The protein contents of the envelope fractions were analysed by SDS-PAGE and coomassie staining. Prior to loading, samples were heated for 5 min at 90˚C, a temperature which is not sufficient to fully denature OmpA (folded OmpA, fOmpA). The band of BamB overlaps with the band of the major porin unfolded OmpC (uOmpC). (**B**)

*Figure 3 continued on next page*

*Figure 3 continued*

The BW25113 and the derivative Δ*dolP*, Δ*skp*, or Δ*ompA* strains carrying an empty control vector (pCtrl) or pBAM$^{His}$ were serially diluted and spotted onto LB agar supplemented with IPTG as indicated. (**C**) BW25113 cells carrying a control empty vector (pCtrl), or the indicated plasmids for ectopic overproduction of BAM, or subsets of BAM subunits, or OmpA$^{His}$ were serially diluted and spotted onto LB agar containing 400 μM IPTG. The diagrams depict the overproduced proteins. (**D**) BW25113 and derivative Δ*dolP* cells carrying the indicated plasmids for ectopic overproduction of either BamA alone or both DolP$^{His}$ and BamA were serially diluted and spotted onto LB agar supplemented with IPTG as indicated. (**E**) BW25113 cells carrying the indicated plasmids were cultured overnight and streaked onto LB agar containing IPTG and vancomycin as indicated. (**F**) Heat-modifiability of BamA in wild-type and Δ*dolP* cells carrying the indicated plasmids. When the cultures reached the mid-exponential phase, the expression of BamA was induced for 2 hr with 200 μM IPTG. Total cell proteins were incubated at 25°C (Boiling −) or at 99°C (Boiling +), separated by SDS-PAGE and analysed by immunoblotting using the indicated antisera. u, unfolded; f, folded. (**G**) Heat modifiability of the protein contents of the envelope fraction of BW25113 (*dolP*$^+$) or Δ*dolP* cells carrying no vector or transformed with pBamA or pDolP$^{His}$-BamA. Plasmid-borne genes were induced with 200 μM IPTG for 2 hr prior to collecting cells. The envelope fractions were mixed with SDS-PAGE loading buffer, incubated at 25°C (Boiling −) or 99°C (Boiling +) for 10 min, and analysed by SDS-PAGE and coomassie staining. u, unfolded.

The online version of this article includes the following figure supplement(s) for figure 3:

**Figure supplement 1.** The detrimental effect of BAM overproduction is caused by the overaccumulation of BamA in the OM.

by σ$^E$ and its deletion causes sensitivity to vancomycin (*Figure 1—figure supplement 1B*). In contrast to Δ*dolP*, a Δ*skp* strain harbouring the same BAM overproduction plasmid could grow as efficiently as the wild-type reference (*Figure 3B*). Similarly, cells lacking OmpA, which is downregulated by σ$^E$ activation, could tolerate BAM overproduction (*Figure 3B*). These results suggest a specific effect of DolP in supporting cell fitness when BAM is overproduced.

The excess of BamA alone was responsible for the observed growth defect, as the excess of different subsets of BAM subunits that did not include BamA or an excess of OmpA obtained using a similar overproduction plasmid (see also the subsequent description of *Figure 5—figure supplement 3C*) had no detectable effects in our growth tests (*Figure 3C*). The detrimental effect of BAM overproduction was caused by the overaccumulation of BamA in the OM, as the overproduction of an assembly-defective BamA variant, BamA$^{ΔP1}$, that lacks the N-terminal POTRA1 motif and that largely accumulates in the periplasm (*Bennion et al., 2010*), did not impair growth to the same extent (*Figure 3—figure supplement 1A–1C*). Strikingly, the growth defect caused by the overproduction of BamA was fully rescued by the concomitant overproduction of DolP (*Figure 3D*), indicating a dose-dependent positive-fitness effect of DolP. Most importantly, we noticed that a lower induction of BamA expression (50 μM IPTG) did not cause any major growth defect but determined a marked sensitivity of wild-type cells to vancomycin (*Figure 3E*). Even under these conditions, the concomitant overproduction of DolP was beneficial and restored growth, thus revealing that DolP contributes to rescue an OM integrity defect caused by increased BamA levels (*Figure 3E*).

To better understand this phenotype, we analysed the levels of heat-modifiable (properly folded) versus non-heat-modifiable (improperly folded) BamA in cells overproducing equal amounts of BamA and lacking or expressing different levels of DolP. A large fraction of overproduced BamA in wild-type or Δ*dolP* cells was non-heat-modifiable (*Figure 3F*, lanes 1–4). Given that overproduced BamA did not accumulate in the periplasm (*Figure 3—figure supplement 1D*), we deduce that improperly folded BamA is associated with the OM. In contrast to BamA, OmpA that was overproduced with a similar plasmid and the same amount of inducer was quantitatively heat-modifiable (*Figure 3—figure supplement 1E*), indicating that this degree of protein expression did not saturate the OMP biogenesis machinery. We noticed that cells overproducing BamA had reduced OMP levels (*Figure 3G*, lanes 5–8), suggesting that improperly folded BamA may interfere with the OMP biogenesis activity of the endogenous BAM complex. Strikingly, when DolP was concomitantly overproduced with BamA, virtually all BamA was found to be heat-modifiable, suggesting that it could properly fold (*Figure 3F*, lanes 5–8). The overproduction of DolP partially rescued the wild-type level of OMPs (*Figure 3G*, lanes 9–12), indicating some degree of restoration of the BamA function. The fact that the wild-type OMP levels were not fully restored probably owes to the lack of stoichiometric amounts of the BAM lipoproteins compared to the amount of BamA in these cells. Taken together these results suggest that the detrimental effects inherent to incremental BamA expression correlate well with the observed BamA folding defect. Most importantly, DolP can restore proper folding of BamA rescuing OM integrity. We could not detect the non-heat-modifiable form of BamA produced at endogenous levels in a Δ*dolP* genetic background (*Figure 3—figure supplement 1E*).

It is possible that low levels of improperly folded BamA would be promptly degraded (*Narita et al., 2013*), whereas when BamA is produced at higher levels, the larger fraction of improperly folded BamA may not be degraded as efficiently and is thus detected. Overall, these results indicate that DolP supports proper folding and functioning of BamA.

## DolP interacts with BamA assembled in the OM

We wished to investigate whether DolP physically interacts with the BAM complex. In a first set of pull-downs, we exploited the specificity of staphylococcal protein A-IgG binding to investigate a possible BAM–DolP association. A construct encoding C-terminally protein A-tagged DolP was ectopically expressed in Δ*dolP* cells. The envelope of cells expressing DolP$^{ProtA}$ was solubilized using digitonin as main mild-detergent component prior to IgG-affinity chromatography (*Figure 4A*, Coomassie staining). Site-specific enzymatic cleavage of an amino acid linker between DolP and the protein A tag was used for protein elution. Notably, BamA, BamC, BamD, and BamE were immunodetected in the elution fraction of protein A-tagged DolP (*Figure 4A*, lane 3). In contrast, the membrane proteins OmpA and CyoA, and cytosolic RpoB were not detected. Next, BAM$^{ProtA}$ (consisting of wild-type BamABCD and a C-terminally protein A-tagged BamE protein variant) was ectopically overproduced to isolate the BAM complex via IgG-affinity purification (*Figure 4—figure supplement 1A*, Coomassie staining). Along with the BamE bait and other subunits of the BAM complex, DolP was also immunodetected in the elution fraction (*Figure 4—figure supplement 1A*, lane 3). Other proteins of the bacterial envelope (Skp, LamB, OmpA, and F$_1$β of the F$_1$F$_O$ ATP synthase) or cytosolic RpoB were not detected. Taken together, our native pull-down analysis indicates that DolP and BAM have affinity for each other.

To explore whether the central BAM subunit, BamA, is a critical determinant of the BAM–DolP interaction, we performed Ni-affinity purification using the solubilized envelope fraction obtained from cells overproducing BamA and C-terminally polyhistidine-tagged DolP. Under these conditions, BamA was efficiently co-eluted together with DolP$^{His}$, demonstrating that BamA and DolP can interact even in the absence of stoichiometric amounts of the BAM lipoproteins (*Figure 4B*, Coomassie staining). To assess if the interaction of DolP and BAM takes place at the OM, DolP$^{His}$ was overproduced together with the assembly-defective form BamA$^{ΔP1}$. When expressed together with DolP$^{His}$, assembly-defective BamA$^{ΔP1}$ was highly depleted in the corresponding eluate (*Figure 4B*, lane 7), even though BamA$^{ΔP1}$ was only marginally reduced in the crude envelope fraction with respect to wild-type BamA (*Figure 4B*, lane 3). In contrast to BamA$^{ΔP1}$, the BamA$^{ΔP2}$ variant, which is efficiently assembled into the OM (*Figure 3—figure supplement 1C*), was co-eluted to a similar extent as wild-type BamA (*Figure 4B*, lane 8). We conclude that DolP has affinity for OM-assembled BamA.

To verify the proximity of DolP to BamA in living cells, we performed in vivo site-directed photo-crosslinking. A photo-activatable amino acid analog, p-benzoyl-L-phenylalanine (Bpa), was introduced by amber suppression (*Chin et al., 2002*) at 17 distinct positions in DolP, three in the linker between the N-terminal lipid-modified cysteine residue and the first BON domain (BON1), eight in BON1, and six in the second BON domain (BON2). Bpa can crosslink with other amino acids at a distance of 3–4 Å, possibly revealing direct protein–protein interactions. Upon UV irradiation and Ni-affinity purification of DolP, several crosslink products were identified by immunoblotting (*Figure 4C* and *Figure 4—figure supplement 1B*). Using both anti-DolP and anti-BamA antibodies, a crosslink product of approximately 115 kDa was detected with samples containing Bpa at positions V52, V72 and, to a lower extent, V101. Notably, positions V52, V72, and V101 are proximal in the three-dimensional structure of DolP with their side chains exposed on the surface of BON1 and oriented away from BON2 (*Figure 4C*). Thus, these results identify in BON1 a site of interaction of DolP with BamA. In addition, a major crosslink product with an apparent molecular weight of 55 kDa was detected with Bpa at several DolP positions, and most prominently V52, I64, V72, and V101 of BON1. This product is approximately 35 kDa larger than the mass of DolP. When analysed by MALDI-TOF mass spectrometry, tryptic peptides of DolP and OmpA (37 kDa) were identified in the 55 kDa crosslink products obtained with Bpa at position V41 and V52 (*Figure 4—figure supplement 2A–C*). LC-MS/MS analysis further identified a peptide of the C-terminal domain of OmpA, which localizes in the periplasm (*Ishida et al., 2014*), to be crosslinked by Bpa at position V52 of DolP (*Figure 4—figure supplement 2D–2F*). Thus, whereas OmpA could not be detected in native pull-downs of affinity-tagged DolP, it was efficiently crosslinked. Compared to Δ*dolP*, cells lacking OmpA are not susceptible to vancomycin treatment (*Figure 4—figure supplement 1D*) and can tolerate

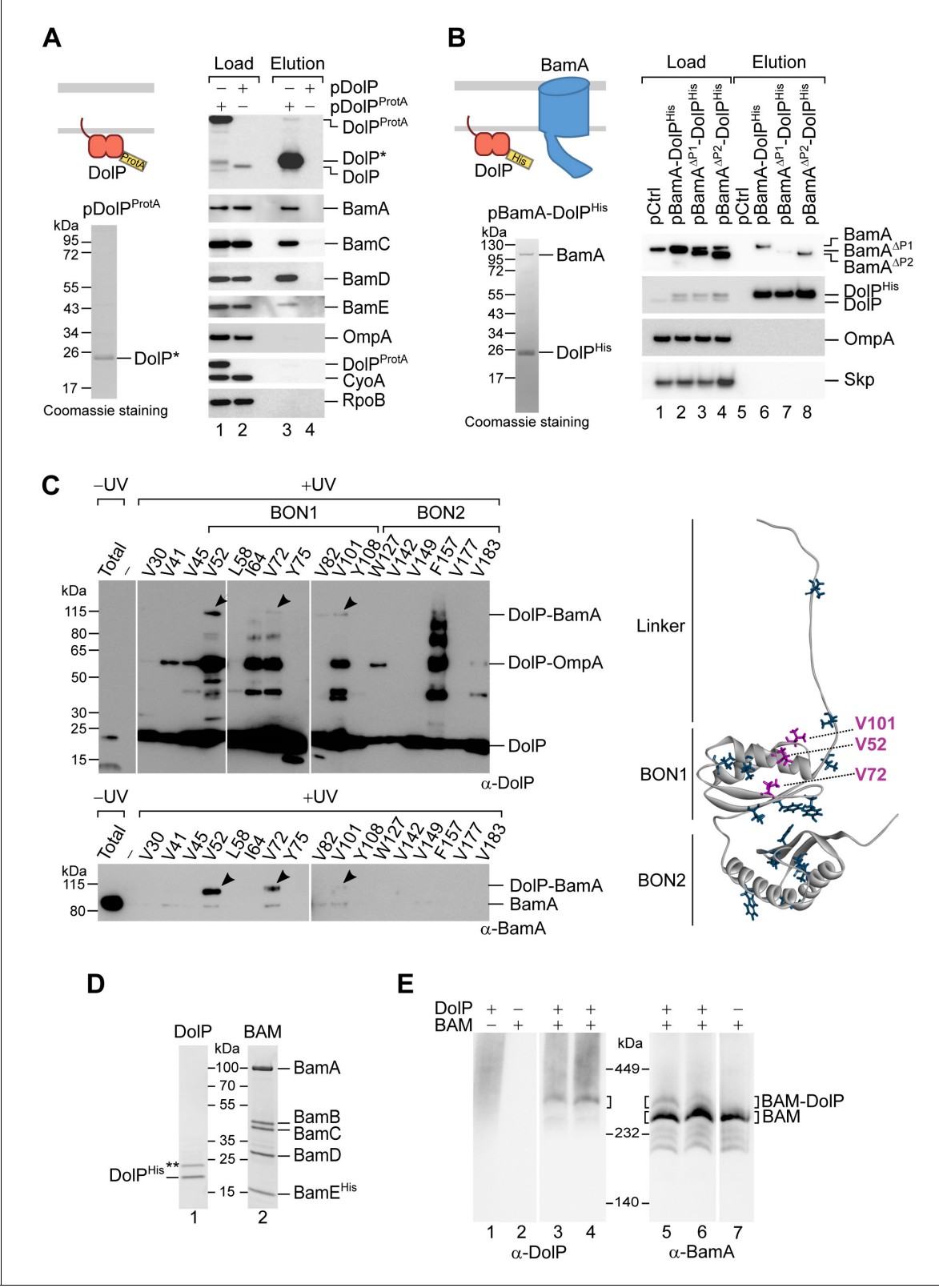

**Figure 4.** DolP associates with the BAM complex via an interaction with BamA. (**A**) The envelope fractions of BW25113 cells carrying the indicated plasmids were solubilized with 1% (w/v) digitonin and 0.1% (w/v) DDM and subjected to IgG affinity purification of protein A-tagged DolP. The load and elution fractions were analysed by SDS-PAGE. The coomassie staining of the elution of protein A-tagged DolP is shown below the diagrams representing the overproduced protein. Blotted proteins from load and elution fractions were detected by immunolabelling using the indicated

*Figure 4 continued on next page*

*Figure 4 continued*

antisera. Load 0.5%; Elution 100%. The asterisk indicates the TEV-digestion product of DolP$^{ProtA}$. (B) The envelope fractions of BW25113 cells carrying the plasmids overproducing His-tagged DolP and the indicated BamA protein variants (deleted of POTRA1 or of POTRA2) were solubilized with 1% (w/v) digitonin and 0.1% (w/v) DDM and subjected to Ni-affinity purification. The load and elution fractions were analysed by SDS-PAGE. The coomassie staining of the elution of His-tagged DolP overproduced together with wild-type BamA is shown below the diagram representing the overproduced proteins. Blotted protein from load and elution fractions were detected by immunolabelling using the indicated antisera. Load 2%; Elution 100%. The amount of BamA co-isolated with DolP$^{His}$ was normalized to the amount of BamA detected in the load fraction. The value obtained for the pBamA-DolP$^{His}$ sample was set to 100%. The average of the relative amounts of co-isolated BamA$^{\Delta P1}$ and BamA$^{\Delta P2}$ are as follows: BamA$^{\Delta P1}$, 16.5% (N = 2; 1st exp. 23.6%; 2nd exp. 9.3%); BamA$^{\Delta P2}$, 81.2% (N = 2; 1st exp. 101.8%; 2nd exp. 60.6%). (C) UV photo-crosslinking of ΔdolP cells transformed with pEVOL-pBpF and pBamA-DolP$^{His}$ harbouring an amber codon at the indicated position of the *dolP* ORF. Upon Ni-affinity chromatography of DolP$^{His}$, eluates obtained from UV irradiated samples were separated by SDS-PAGE and analysed by immunoblotting using the indicated antisera. The total envelope fraction of cells expressing DolP$^{His}$ with Bpa at position V52 (non-irradiated) is shown in the first lane and serves as a reference for the migration of non-crosslinked DolP and BamA. Arrowheads indicate crosslinked products detected with both DolP and BamA antisera. Analysis of eluates obtained from non-irradiated samples are shown in *Figure 4—figure supplement 1B*. The amino acid residues replaced with Bpa are indicated on the structure of DolP, PDB: 7A2D (*Bryant et al., 2020*). In purple are the positions crosslinked to BamA. (D) The envelope fraction of BW25113 cells overproducing DolP$^{His}$ or the BAM complex containing C-terminally His-tagged BamE was subjected to protein extraction with 1% (w/v) DDM, Ni-affinity purification, and gel filtration chromatography. The elution fractions were analysed by SDS-PAGE and coomassie staining. The double asterisk indicates a contaminant protein in the elution of DolP. (E) Roughly equimolar quantities of purified His-tagged BAM complex and DolP were incubated alone for 1 hr at 4°C (lanes 1, 2, and 7), or together for 1 hr at 4°C (lanes 3 and 6) or for 30 min at 25°C (lanes 4 and 5), prior to blue native-PAGE and immunoblotting using the indicated antisera.

The online version of this article includes the following figure supplement(s) for figure 4:

**Figure supplement 1.** Analysis of the DolP–BamA interaction.

**Figure supplement 2.** Mass spectrometry analyses of the DolP–OmpA crosslink product.

the overproduction of BamA (*Figure 3B*), suggesting that OmpA is not required for DolP function. Finally, a series of crosslink products with molecular weights approximately two to six times the mass of DolP were detected with anti-DolP antibodies when Bpa was introduced at position F157 (*Figure 4C*), suggesting that DolP can form oligomers.

In seeking a detergent that would interfere with the interaction of BAM and DolP, and allow their purification as separate components, we solubilized the envelope fraction with increasing amounts of n-dodecyl β-D-maltoside (DDM), a detergent previously used to isolate the native BAM complex (*Roman-Hernandez et al., 2014*). At concentrations of DDM between 0.3% (w/v) and 1% (w/v), we observed a drastic reduction in the amounts of BAM subunits that were co-eluted with DolP$^{His}$ (*Figure 4—figure supplement 1C*), indicating that the BAM–DolP interaction is sensitive to DDM. We thus used 1% (w/v) DDM to extract and purify His-tagged DolP or His-tagged BAM as separate components (*Figure 4D*). When analysed by blue native-PAGE and immunoblotting, purified DolP gave rise to a diffused signal at around 450 kDa (*Figure 4E*, lane 1), suggesting a dynamic multimeric organization of this protein. Purified BAM migrated as expected at 250 kDa (*Figure 4E*, lane 7). When roughly equimolar amounts of purified BAM and DolP were pre-incubated in the presence of a low DDM concentration and subsequently resolved by blue native-PAGE, a complex with an apparent molecular weight higher than that of the BAM complex was detected with both anti-BamA- and anti-DolP-specific antibodies (*Figure 4E*, lanes 3–6), suggesting that DolP can associate with the penta-subunit BAM complex. Taken together our results demonstrate that DolP can interact with the BAM complex, making direct contacts with OM-assembled BamA.

## BamA overaccumulation in the OM reduces DolP mid-cell localization

In light of our observation that DolP interacts with BAM, we asked whether the envelope localization patterns of DolP and BAM are reciprocally linked. First, we monitored the effect of DolP expression on the localization of the chromosomally encoded BamD$^{mCherry}$ subunit of the BAM complex. This protein generated a fluorescence signal throughout the envelope that was not affected by the lack or the overproduction of DolP (*Figure 5—figure supplement 1*). Next, we checked the effect on DolP localization of BAM overaccumulation in the OM. DolP associates with the OM and accumulates at mid-cell during a late step of cell division (*Tsang et al., 2017*). To monitor the localization of DolP, we used a strain harbouring a chromosomal *dolP-gfp* fusion (*Figure 1A*). The localization of the DolP$^{GFP}$ fusion protein (*Figure 5—figure supplement 2A*) was analysed concomitantly with two other chromosomally encoded markers of the division septum, ZipA$^{mCherry}$ or NlpD$^{mCherry}$. ZipA is

involved in an early step of divisome assembly and accumulates at division sites before, as well as, during envelope constriction (*Figure 5—figure supplement 2B*; *Hale and de Boer, 1997*). Instead, NlpD is a late marker of cell division involved in the activation of AmiC and accumulates at septa that are already undergoing constriction (*Figure 5—figure supplement 2C*; *Uehara et al., 2009*; *Uehara et al., 2010*). DolP$^{GFP}$ accumulated at mid-cell sites where the envelope appeared invaginated, showing a localization pattern similar to that of NlpD$^{mCherry}$ (*Figure 5—figure supplement 2C*; *Tsang et al., 2017*). We investigated the effect of short-lived (1 hr) BAM overproduction on DolP$^{GFP}$ localization. Strikingly, we found that BAM overproduction depleted DolP$^{GFP}$ from mid-cell sites (*Figure 5A*, plot, and *Figure 5—figure supplement 3A*, left). In contrast, no obvious effects on cell division nor on mid-cell recruitment of ZipA$^{mCherry}$ and NlpD$^{mCherry}$ were observed (*Figure 5— figure supplement 3B*). The overproduction of BamA alone was sufficient to alter the distribution of the DolP$^{GFP}$ fluorescence signal in constricting cells, with its intensity being reduced at constriction sites but enhanced at decentred positions along the cell axis (*Figure 5—figure supplement 3A*). In contrast, the overproduction of only the four BAM lipoproteins (*Figure 5—figure supplement 3A*, right) as well as the overproduction of OmpA (*Figure 5—figure supplement 3C*) had no obvious effects on DolP mid-cell localization.

As BAM catalyses OMP assembly, we asked whether this activity interferes with DolP mid-cell localization. To address this question, we made use of an inactive BamA mutant form (*Figure 5—figure supplement 4A*) harbouring a polyhistidine peptide extension at its C-terminal β-strand (*Hartmann et al., 2018*). Similar to the overaccumulation of the BAM complex or BamA, the overaccumulation of BamA$^{His}$ interfered with DolP mid-cell localization (*Figure 5—figure supplement 4B and C*), without affecting ZipA$^{mCherry}$ and NlpD$^{mCherry}$ (*Figure 5—figure supplement 3D*), indicating that the cellular localization of DolP does not depend on the OMP-assembly activity of BamA. In contrast, the ability of BamA to assemble into the OM was a critical determinant of the observed septal depletion of DolP. In fact, the periplasm-accumulating BamA$^{ΔP1/His}$ variant (*Figure 5—figure supplement 4D*) did not impair DolP$^{GFP}$ mid-cell localization (*Figure 5—figure supplement 4B*, centre), whereas the OM-overaccumulating BamA$^{ΔP2/His}$ did (*Figure 5—figure supplement 4B*, right, 4C and 4D). Taken together, these results suggest that the overaccumulation of BamA in the OM interferes with the recruitment of DolP at mid-cell sites.

## DolP mid-cell localization is impaired under envelope stress conditions

Given that DolP is critical for fitness under envelope stress conditions, we wished to know whether envelope stress would influence the localization of DolP$^{GFP}$. To this end, first we analysed the localization of DolP$^{GFP}$ in strains lacking either the OMP chaperone SurA or the lipoprotein BamB. Both Δ*surA* and Δ*bamB* strains are defective in OMP biogenesis and produce higher levels of BAM complex due to activation of the σ$^E$ response (*Charlson et al., 2006*; *Rouvière and Gross, 1996*; *Vertommen et al., 2009*; *Wu et al., 2005*). Importantly, the frequency of mid-cell labelling by DolP$^{GFP}$ was reduced in Δ*surA* cells both in minimal (*Figure 5—figure supplement 5A*, centre) and LB (*Figure 5B*) culture media. In contrast, lack of SurA did not affect septal recruitment of the late cell division marker NlpD (*Figure 5—figure supplement 5B*). The analysis of the fluorescence plot profiles of constricted cells clearly showed a marked reduction of the DolP$^{GFP}$ signal at mid-cell sites and higher fluorescence levels at decentred positions along the cell axis (*Figure 5B*, right plot). As for the Δ*surA* strain, DolP$^{GFP}$ accumulated at the mid-cell with a lower frequency when *bamB* was deleted (*Figure 5—figure supplement 5A*, bottom). Together, these results indicate that, during envelope stress, DolP is depleted at mid-cell sites.

Because the levels of OmpA and OmpC are reduced under envelope stress conditions, we investigated if these OMPs are required for the mid-cell localization of DolP. In both *ompA* or *ompC* deletion strains, we observed marked DolP$^{GFP}$ intensities at cell constriction sites, indicating that neither OmpA nor OmpC are crucial for DolP mid-cell localization (*Figure 5C* and *Figure 5—figure supplement 5C*). Nevertheless, the plot collective profile of DolP$^{GFP}$ in Δ*ompA* showed a marginal reduction of fluorescence intensity at the mid-cell and a similarly small increment at non-septal positions. Like BamA, OmpA was crosslinked with DolP at position V52. By monitoring this crosslink reaction, we found that the DolP-BamA association is enhanced in the absence of OmpA (*Figure 5D*), suggesting that OmpA competes with BamA for an interaction with DolP. Hence, the depletion of OmpA in stressed cells might favour the interaction of DolP with BamA.

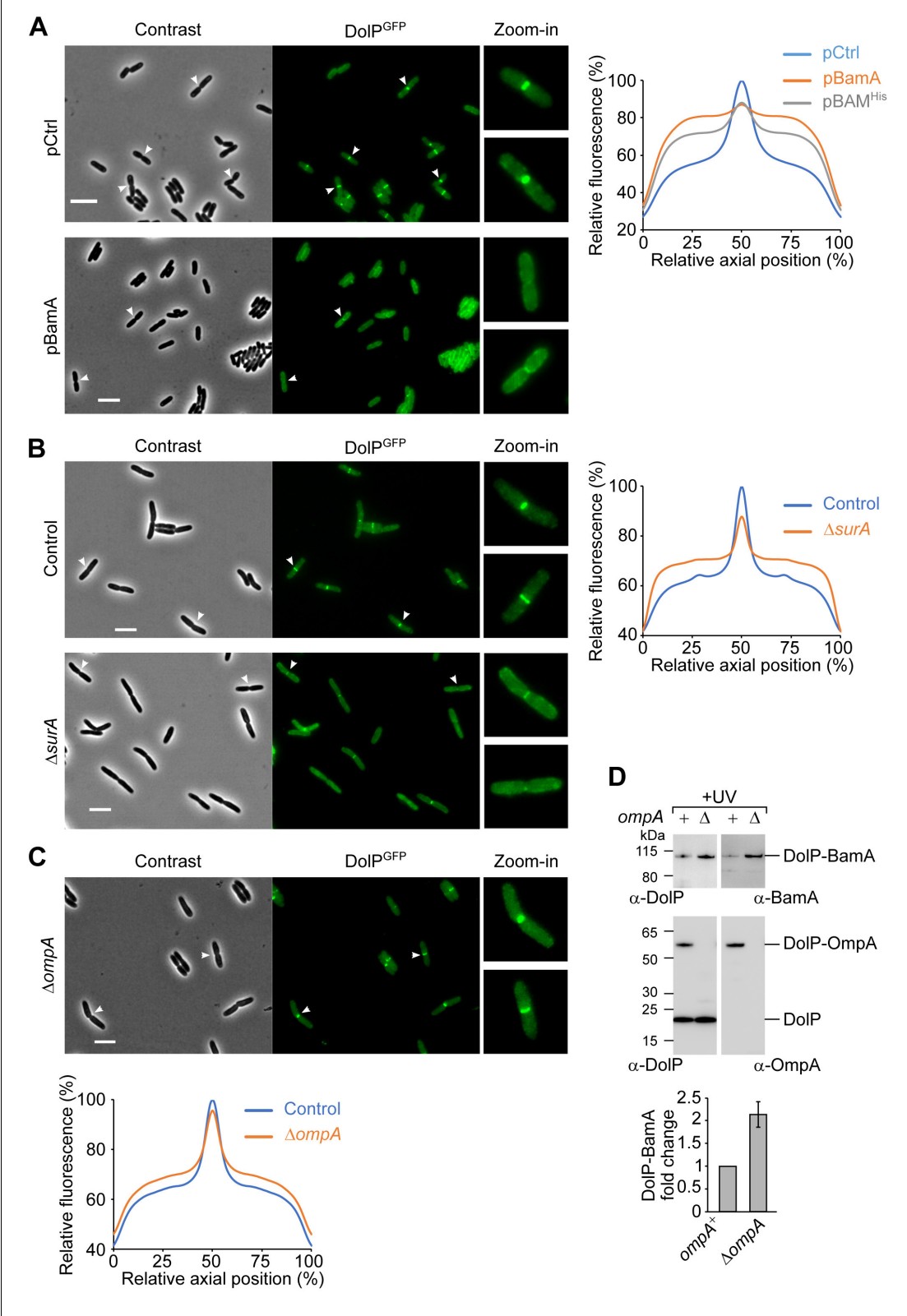

**Figure 5.** BamA overaccumulation in the OM and envelope stress interfere with the mid-cell localization of DolP. (**A**) Overnight cultures of BW25113 cells harbouring the chromosomal fusion *dolP-gfp* and transformed with either pCtrl (empty vector) or pBamA were freshly diluted in minimal M9 medium, incubated at 30°C until $OD_{600}$ = 0.1 and supplemented with 400 µM IPTG for 1 hr. Cell samples were visualized on 1% (w/v) agarose pads by phase contrast and fluorescence microscopy. Arrowheads indicate envelope constriction sites between forming daughter cells. Bar = 5 µm. The

*Figure 5 continued on next page*

*Figure 5 continued*

collective profiles of fluorescence distribution versus the relative position along the cell axis were plotted: pCtrl, blue; pBamA, orange; pBAM$^{His}$, grey (images of cells transformed with pBAM$^{His}$ are shown in **Figure 5—figure supplement 3A**). Only cells with a constriction (N = 361, pCtrl; N = 187, pBamA; N = 187, pBAM$^{His}$) were taken into account for the collective profile plots. Fluorescence intensities were normalized to the mid-cell value obtained for the control sample. (B) Overnight cultures of BW25113 (control) or Δ*surA* derivative cells carrying the *dolP-gfp* chromosomal fusion were freshly diluted in LB medium and incubated at 30°C until OD$_{600}$ = 0.3. Cell samples were visualized as in (A). Bar = 5 μm. The collective profiles of fluorescence distribution versus the relative position along the cell axis is shown for Δ*surA* cells (orange) and *surA*$^+$ control cells (blue). Only cells with a constriction (N = 318, Control; N = 320, Δ*surA*) were taken into account for the collective profile plots. Fluorescence intensities were normalized to the mid-cell value obtained for the control sample. (C) Overnight cultures of Δ*ompA* cells carrying the *dolP-gfp* chromosomal fusion were cultured and visualized as in (B). Bar = 5 μm. The collective profiles of fluorescence distribution versus the relative position along the cell axis is shown for Δ*ompA* cells (orange) and an *ompA*$^+$ (control) strain that was cultured and visualized in a parallel experiment (blue). Only cells with a constriction (N = 287, Control; N = 193, Δ*ompA*) were taken into account for the collective profile plots. Fluorescence intensities were normalized to the mid-cell value obtained for the control sample. (D) UV photo-crosslinking of Δ*dolP* and Δ*dolP* Δ*ompA* cells transformed with pBamA-DolP$^{His}$ harbouring an amber codon at position V52 of the *dolP* ORF. Signals obtained with the anti-BamA antiserum were quantified and showed in the histogram. The amount of DolP-BamA crosslink product obtained with samples lacking OmpA is expressed as fold change of the amount of the same product obtained in samples expressing OmpA. Data are reported as mean ± SEM (N = 3).

The online version of this article includes the following figure supplement(s) for figure 5:

**Figure supplement 1.** Effect of the lack or the overproduction of DolP on BAM localization.
**Figure supplement 2.** DolP$^{GFP}$, NlpD$^{mCherry}$, and ZipA$^{mCherry}$ mid-cell localization patterns.
**Figure supplement 3.** Overproduction of BAM influences septal recruitment of DolP$^{GFP}$ but not NlpD$^{mCherry}$ or ZipA$^{mCherry}$.
**Figure supplement 4.** BamA overaccumulation in the OM impairs mid-cell localization of DolP$^{GFP}$.
**Figure supplement 5.** Envelope stress influences the localization of DolP$^{GFP}$.

## Discussion

Upon envelope stress, the BAM complex and other OM biogenesis factors are upregulated to meet the cellular demand for OMP assembly. The role of DolP, which is also upregulated by the envelope stress response, was unclear. In our study, we uncover that DolP supports cell fitness under envelope stress conditions and we demonstrate that the BAM complex, in particular its central catalytic sub-unit BamA, is a direct target of such fitness function.

First we have shown that in cells that lack BamB and undergo envelope stress, DolP is important to maintain the levels of BamA. Next, we have exploited the observation that, when overproduced, BamA impairs OM integrity, causing a detrimental effect dependent on its accumulation in the OM. Under these conditions, a significant portion of membrane-embedded BamA is improperly folded and OMP biogenesis is reduced. We have shown that an increment of DolP expression rescues proper folding of BamA and, to some extent, efficient OMP biogenesis. Finally, we have demonstrated that DolP directly interacts with the BAM complex in the OM, making contacts to BamA. Taken together, these results strongly suggest that DolP contributes to preserve the OM integrity by supporting the function of BamA, thus shedding some light on how DolP copes with envelope stress.

In addition to promoting efficient OMP biogenesis, proper folding of BamA may be necessary to regulate the dynamics of this protein in the OM. As part of the mechanism by which BAM functions in OMP biogenesis, BamA is predicted to interfere with the organization of the surrounding lipid bilayer and to generate an energetically favourable environment for the insertion of nascent OMPs in the OM (*Fleming, 2015*; *Horne et al., 2020*). BamA-mediated destabilization of a lipid bilayer was shown by molecular dynamics simulations, as well as by reconstituting both BamA into proteoliposomes and the BAM complex into nanodiscs (*Iadanza et al., 2020*; *Noinaj et al., 2013*; *Sinnige et al., 2014*). Furthermore, when reconstituted into a lipid bilayer, BamA can form pores characterized by variable conductance (*Stegmeier and Andersen, 2006*). With an improperly folded conformation, these features of BamA may be uncontrolled and potentially detrimental for OM integrity. The finding that DolP supports proper folding of BamA provides an explanation as to how DolP helps preserving OM integrity. This role of DolP is reminiscent of the chaperone function attributed to a different dual BON-domain protein, OsmY, which promotes the folding of a specific sub-class of OMPs (*Yan et al., 2019*). We speculate that, by associating with BamA, DolP directly contributes to its folding in the OM. The evidence that DolP associates but does not form a stoichiometric complex with BAM is consistent with the hypothesis that DolP may transiently act on BamA

similar to a chaperone. Notably, although the chaperone SurA is not required for BamA assembly in the OM, it was shown that in the absence of this factor a portion of BamA is proteolytically degraded (*Bennion et al., 2010*; *Narita et al., 2013*). It is thus tempting to explain the quasi-lethal phenotype of a double *surA* and *dolP* deletion strain (*Onufryk et al., 2005*) with a scenario where at least SurA or DolP must be expressed to maintain BamA in a properly folded conformation. Further studies will be warranted to determine in detail the molecular bases of how DolP helps preserving the proper folding of BamA in the OM.

The observation that DolP binds anionic phospholipids, such as phosphatidylglycerol and cardiolipin (*Bryant et al., 2020*), is particularly interesting in the contest of the BAM–DolP interaction. Phospholipid binding is mediated by residues in BON2 of DolP, away from to the site of interaction with BamA that we have identified in BON1. Conceivably, the binding of BamA may not interfere with the ability of DolP to interact with phospholipids. However, whether phospholipid binding by DolP plays a role in supporting proper folding and activity of BamA remains to be determined. In an in vitro experimental set-up, the OMP-assembly activity of BamA is only marginally dependent on the surrounding lipid content (*Hussain and Bernstein, 2018*). It is intriguing however that amino acid residues of the BamA POTRA domains can make contact with the lipid head-groups on the periplasmic surface of the OM and that BamE interacts with an anionic phospholipid (*Fleming et al., 2016*; *Knowles et al., 2011*). Thus, two non-mutually exclusive scenarios should be considered: (i) enriching the BAM sites with negatively charged phospholipids may be part of the mechanism by which DolP contributes to maintain BamA in a properly folded conformation; (ii) DolP may interact with phospholipids in proximity of the BAM complex and form a structure that helps preserving the integrity of these sites. Of note, we have obtained some evidence that DolP can form oligomers, but it remains to be established whether these DolP structures form in proximity of the BAM complex.

Whereas we have shown that DolP restores an OM integrity defect inherent to BamA expression, we cannot exclude that also other OM features determined by the activation of the envelope stress response require the activity of DolP. In any event, a key finding of our study is that DolP mid-cell localization is sensitive to envelope stress conditions. During envelope stress, the OM undergoes a significant alteration of its protein composition, with a marked downregulation of porins and OmpA (*Rhodius et al., 2006*). OMPs are largely arranged in clusters (*Gunasinghe et al., 2018*; *Jarosławski et al., 2009*; *Rassam et al., 2015*) embedded by highly organized LPS molecules in the external leaflet of the OM (*Nikaido, 2003*) and the rigidity of their β-barrel structures contributes to the mechanical stiffness of the OM (*Lessen et al., 2018*). Downregulation targets of σ$^E$ also include the lipoproteins Pal and Lpp (*Gogol et al., 2011*; *Guo et al., 2014*), which are critical for OM integrity (*Asmar and Collet, 2018*; *Cascales et al., 2002*). Importantly, we have shown that the overaccumulation of BAM in the OM influences DolP localization, suggesting that during stress the upregulation of BamA contributes to deplete DolP from cell constriction sites. We have also obtained evidence that OmpA competes with BamA for an interaction with DolP. A role of OmpA in buffering the function of an envelope stress factor has been reported (*Dekoninck et al., 2020*). We propose that during envelope stress, the depletion of OmpA might enhance the interaction of DolP with BamA.

Distinct biogenesis and surveillance pathways are required to maintain the protective function of the multi-layered envelope of Gram-negative bacteria (*Egan et al., 2020*). The hits of our CRISPRi synthetic-defect screen in Δ*dolP* cells include mainly genes involved in efficient transport and assembly of OMPs or in the activation of the σ$^E$-mediated envelope stress response, which is consistent with the notion that DolP supports the high demand for OMP biogenesis during stress. In addition, genes involved in the activation of the AmiA pathway of septal peptidoglycan splitting were identified. This result is in line with the conclusions of a previous study implicating DolP in the regulation of the NlpD-mediated activation of AmiC (*Tsang et al., 2017*). Cells lacking both NlpD and AmiC have reduced OM integrity, which contributes to explain the vancomycin sensitivity of Δ*dolP* cells (*Tsang et al., 2017*). Intriguingly, our observation that mid-cell localization of DolP is reduced under envelope stress conditions points to a possible role of DolP in linking envelope stress to septal peptidoglycan hydrolysis. Reduced levels of DolP at mid-cell sites, and thus impaired AmiC activation (*Tsang et al., 2017*), could play an important role in coping with envelope stress, for instance by regulating the window of time available to restore efficient OMP biogenesis prior to completing the formation of the new poles in the cell offspring.

In summary, our results reveal an unprecedented function for DolP in supporting the correct folding of BamA, providing an explanation as to how DolP promotes OM integrity and why this factor is upregulated during envelope stress. The identified role of DolP in supporting the BamA function represents a potentially exploitable target in the development of alternative antibacterial therapies. The re-localization of DolP during stress points to a mechanistic link between activation of the envelope stress response and a late step of cell division that will be interesting to investigate in future studies.

## Materials and methods

### Bacterial strains and growth conditions

All *E. coli* strains used in this study are listed in *Supplementary file 1*. Strains newly generated for this study derive from BW25113 [Δ(*araD-araB*)567 Δ(*rhaD-rhaB*)568 Δ*lacZ4787* (::rrnB-3) *hsdR514 rph-1*] (*Grenier et al., 2014*) or MG1655 (F⁻ λ⁻ *ilvG⁻ rfb-50 rph-1*) (*Blattner et al., 1997*). Deletions of *dolP, rseA, surA, bamB, degP, skp, ompA,* or *ompC* were achieved by P1 transduction of the Δ*dolP::kan*, Δ*rseA::kan*, Δ*surA::kan*, Δ*bamB::kan*, Δ*degP::kan*, Δ*skp::kan*, Δ*ompA::kan*, or Δ*ompC:: kan* alleles, respectively, obtained from the corresponding Keio collection strains (*Baba et al., 2006*). BW25113 derivative strains harbouring chromosomal fusions of constructs encoding superfolder GFP downstream of *dolP* or mCherry downstream of *nlpD, zipA,* and *bamD* were obtained by λ-red recombination as previously described (*Datsenko and Wanner, 2000*). Briefly, a kanamycin-resistance cassette was amplified from plasmid pKD4 using oligonucleotides carrying extensions of approximately 50 nucleotides homologous to regions immediately upstream or downstream the stop codon of the interested genes. *Dpn*I-digested and purified PCR products were electroporated into the BW25113 or derivative strains. Recombinant clones were selected at 37°C on LB agar plates containing kanamycin. When necessary, the kanamycin-resistance cassette inserted into a mutated locus (gene deletion or fusion) was removed upon transformation with the heat-curable plasmid pCP20 (*Datsenko and Wanner, 2000*). The MG1655 derivative strain LC-E75, harbouring a dCas9-encoding construct under the control of the P$_{tet}$ promoter, has been described (*Cui et al., 2018*). Cells were cultured in home-made lysogeny broth (LB) medium (1% (w/v) tryptone, 0.5% (w/v) yeast extract, 5 mg/ml (NaCl), commercially available Miller LB Broth (Sigma) or M9 minimal medium containing M9 salts (33.7 mM Na$_2$HPO$_4$, 22 mM KH$_2$PO$_4$, 8.55 mM NaCl, 9.35 mM NH$_4$Cl) and supplemented with 0.2% w/v glycerol and all the amino acids. Antibiotics were used at the following concentrations: ampicillin 100 μg/ml, kanamycin 50 μg/ml, and vancomycin 60 μg/ml. For spot tests, cells were cultured to mid-log phase, washed with M9 salts, and serially diluted in ice-cold M9 salts prior to spotting on agar plates.

### Plasmid construction

All plasmids used in this study are listed in *Supplementary file 2*. Plasmids for the ectopic expression of BAM subunits, DolP, or OmpA are derived from a pTrc99a vector. The plasmid pBAM$^{His}$ (pJH114), which harbours a P$_{trc}$ promoter followed by the sequences of the *E. coli* K12 *bamA* ribosome-binding site, the *bamABCDE* open reading frames, and an octahistidine tag fused downstream of *bamE*, was described (*Roman-Hernandez et al., 2014*). The region of pBAM$^{His}$ comprising the segment that spans from the *bamA* start codon to the *bamE* stop codon was deleted by site-directed mutagenesis, generating pCtrl. Plasmids pBamA$^{His}$ was generated by restriction-free cloning, inserting the *bamA* ORF without its stop codon downstream of the P$_{trc}$ promoter and upstream of the octahistidine encoding region in pCtrl. pBamA$^{His}$ was subjected to site-directed mutagenesis to generate pBamA, encoding wild-type, non-tagged BamA. The *dolP* ORF amplified from the BW25113 genomic DNA was used to replace the *bamABCDE* ORFs in pJH114 by restriction-free cloning, generating pDolP$^{His}$. The plasmid pBamA-DolP$^{His}$ was generated by restriction-free cloning of the *bamA* ORF between the P$_{trc}$ promoter and *dolP* in pDolP$^{His}$. Where indicated, site-directed mutagenesis on pBamA-DolP$^{His}$ or pDolP$^{His}$ was used to replace specific *dolP* codons with an amber codon. pDolP$^{His}$-BamA was built in two steps starting from pDolP$^{His}$. First, the sequence of the *E. coli* K12 *bamA* ribosome-binding site was deleted positioning the *dolP* ORF eight nucleotides downstream of the ribosome-binding site of the pTrc99a multiple cloning site. The resulting plasmid was

then used to insert a segment of pJH114 containing the entire *bamA* ORF including the *E. coli bamA* ribosome-binding site.

Site-directed mutagenesis was conducted on pBAM$^{His}$ (pJH114) to obtain pBamACDE$^{His}$, pBamABDE$^{His}$, pBamBCDE$^{His}$, and pBamCDE$^{His}$. A sequence encoding the tobacco etch virus protease cleavage site (TEV site) followed by a tandem Protein A tag was amplified from pYM10 (*Knop et al., 1999*) and fused by restriction-free cloning with the last codon of the *bamE* gene in pBAM$^{His}$ to generate pBAM$^{ProtA}$. A stop codon was introduced downstream of the *bamE* last codon to generate pBAM. Plasmids encoding the ΔP1 and ΔP2 BamA variant were obtained by site-directed mutagenesis deleting the portion of *bamA* ORFs corresponding to residues E22-K89 or P92-G172, respectively. The TEV site and the tandem Protein A construct amplified from pYM10 were inserted by restriction-free cloning downstream of the *dolP* last codon in pDolP$^{His}$, generating pDolP$^{ProtA}$. pDolP was derived from pDolP$^{ProtA}$ using site-directed mutagenesis to introduce a stop codon immediately downstream of the *dolP* ORF. The *ompA* ORF was amplified from the BW25113 genomic DNA and inserted by restriction-free cloning between P$_{trc}$ and the His-tag encoding construct of pCtrl to generate pOmpA$^{His}$. The sgRNAs plasmids are derived from psgRNAcos (*Cui et al., 2018*). To generate sgRNA-encoding plasmids the DNA sequences AGCTGCACCTGCTGCGAATA (*bamD* sgRNA, plasmid pCAT187), GTAAACCACTCGCTCCAGAG (*bamE* sgRNA, plasmid pCAT189), CTCATCCGCG TGGGCGGAAA (*envC* sgRNA, plasmid pCAT191), and CTGAGCCGCCGACCGATTTA (*ftsX* sgRNA, pCAT193) were inserted into a *Bsa*I site of the psgRNAcos.

## CRISPRi screen and data analysis

Strain LC-E75 (*dolP*$^{+}$) and its Δ*dolP* derivative were transformed with the EcoWG1 library which contains five guides per gene as previously described (*Calvo-Villamañán et al., 2020*). After culturing pooled transformant cells in LB at 37°C to early exponential phase (optical density at 600 nm [OD$_{600}$] =0.2), a sample was withdrawn for plasmid isolation (t$_{start}$). Subsequently, cultures were supplemented with 1 µM anhydrotetracycline (aTc) to induce dCas9 expression and further incubated at 37°C. When cultures reached an OD$_{600}$ of 2 they were diluted 1:100 into LB supplemented with 1 µM aTc and incubated at the same temperature until an OD$_{600}$ of 2. This step was repeated one more time prior to withdrawing a sample for isolation of plasmid DNA (t$_{end}$). Sequencing indexes were used to assign reads to each sample. Illumina sequencing samples were prepared and analysed as previously described (*Cui et al., 2018*). Briefly, a two-step PCR was performed with Phusion polymerase (Thermo Scientific) using indexed primers. The first PCR adds the first index and the second PCR adds the second index and flow-cell attachment sequences. Pooled PCR products were gel-purified. Sequencing was performed on a NextSeq550 machine (Illumina). The total number of reads obtained for each sample was used to normalize raw reads by sample size. Replicates were pooled to increase depth before another normalization by sample size. Guides with less than 100 normalized read counts in initial time points were discarded. For each screen, sgRNA fitness was calculated as the log2-transformed ratio of normalized reads counts between the final and the initial time point:

$$log2FC = log2\left(\frac{Normalized\,reads_{final} + 1}{Normalized\,reads_{initial} + 1}\right)$$

For each sample, log2FC values were centred by subtracting the median log2FC of non-targeting control guides. We then calculated for each sgRNA the difference of log2FC value between the Δ*dolP* screen and the *dolP*$^{+}$ screen. Guides were ranked from the lowest negative values (negative fitness effect in Δ*dolP* compared to *dolP*$^{+}$) to the highest positive values (positive fitness effect in Δ*dolP* compared to *dolP*$^{+}$) and the significance of the interaction between *dolP* and each gene was evaluated by performing a minimum hypergeometric (mHG) test on the ranked list for each gene using the mHG R package (v. 1.1) (*McLeay and Bailey, 2010*). False-discovery rate (FDR) was used to correct p-values for multiple testing. For each gene, the median difference of log2FC between Δ*dolP* and *dolP*$^{+}$ screens was used as a measure of the genetic interaction.

## Cell fractionation

To prepare whole-cell lysates, cells were cultured to early exponential phase (OD$_{600}$ = 0.2–0.3) in LB medium at 37°C and collected. Where indicated, IPTG was added 1 or 2 hr prior to cell collection, as indicated. Cells were pelleted by centrifugation, washed once with M9 salt, and lysed with Laemmli

Sample Buffer (Bio-Rad) (69 mM Tris-HCl, pH 6.8, 11.1% [v/v] glycerol, 1.1% [w/v] lithium dodecyl sulphate [LDS], 0.005% [w/v] bromophenol blue, supplemented with 357 mM β-mercaptoethanol and 2 mM phenylmethylsulfonyl fluoride [PMSF]). The whole-cell lysates were heat-denatured at 98°C for 5 min prior SDS-PAGE analysis.

To obtain spheroplasts cells were cultured to early exponential phase, collected by centrifugation, resuspended in 33 mM Tris-HCl, pH 8, 40% (w/v) sucrose to an $OD_{600}$ of 1. The cell suspension was then supplemented with 0.1 mg/ml lysozyme (Sigma), 2 mM EDTA, and incubated on ice for 20 min to induce lysis. After addition of 10 mM $MgSO_4$, the spheroplast fraction was collected by centrifugation at 16,000 × $g$. The supernatant was further centrifuged at 100,000 × $g$ to remove any residual membrane fraction, which was discarded. The obtained soluble (periplasm) fraction was subjected to protein precipitation by adding 10% (w/v) trichloroacetic acid (TCA). TCA precipitates were solubilized in Laemmli Sample Buffer (Bio-Rad) prior to SDS-PAGE analysis. A similar procedure, with a cell resuspension buffer lacking sucrose, was used to lyse cells and obtain the membrane and soluble fractions.

The crude envelope fractions directly analysed by SDS-PAGE or used for native affinity purification of affinity tagged BAM complex or DolP were prepared from cells that were cultured in LB until early exponential phase and, where indicated, supplemented with 400 µM IPTG for 1 or 2 hr (as reported in the figure legends) to induce ectopic protein expression. Cells were collected by centrifugation at 6000 × $g$ at 4°C, resuspended in 20 mM Tris-HCl pH 8, and mechanically disrupted using a Cell Disruptor (Constant Systems LTD) set to 0.82 kPa. The obtained cell lysate fractions were clarified by centrifugation at 6000 × $g$ and 4°C. The supernatant was then subjected to ultracentrifugation at 100,000 × $g$ at 4°C to collect the envelope fraction.

## Protein heat-modifiability

Whole-cell lysates or the crude envelope fractions diluted in Laemmli Sample Buffer were incubated at different temperatures (as indicated in figures and figure legends) prior to analysis by SDS-PAGE and immunoblotting. For the analysis of BamA heat-modifiability, samples were incubated either at 25°C or at 99°C for 10 min and gel electrophoresis was conducted at 4°C.

## Isolation of native protein complexes by IgG- or nickel-affinity chromatography

The envelope fraction was resuspended at a concentration of approximately 10 mg/ml in solubilization buffer (20 mM Tris-HCl pH 7.4, 100 mM NaCl, 0.1 mM EDTA, 2 mM PMSF) supplemented with EDTA-free protease inhibitor cocktail (Roche), and 1.1% (w/v) of a mild detergent component corresponding to digitonin (Merck) and DDM (Merck) as indicated in figure legends. To facilitate extraction of membrane proteins, samples were subjected to mild agitation for 1 hr at 4°C. Insoluble material was removed by centrifugation at 16,000 × $g$ at 4°C. To perform IgG affinity purification, membrane-extracted proteins were incubated for 1.5 hr at 4°C with purified human IgG (Sigma) that had been previously coupled with CNBr-activated Sepharose beads (GE Healthcare). After extensive washes of the resin with solubilization buffer containing 0.3% (w/v) digitonin and 0.03% (w/v) DDM, bound proteins were eluted by incubation with AcTEV protease (ThermoFisher) overnight at 4°C under mild agitation. To perform nickel (Ni)-affinity purification, membrane-extracted proteins were supplemented with 20 mM imidazole and incubated with Protino Ni-NTA agarose beads (Macherey-Nagel) for 1 hr at 4°C. After extensive washes of the resin with solubilization buffer supplemented with 50 mM imidazole, 0.3% (w/v) digitonin, 0.03% (w/v) DDM, and the EDTA-free protease inhibitor cocktail (Roche), bound proteins were eluted using the same buffer supplemented with 500 mM imidazole.

## In vitro reconstitution of the BAM–DolP interaction and BN-PAGE analysis

Envelope fractions were obtained from cells carrying pBAM[His] or pDolP[His] and cultured until early exponential phase in LB medium at 37°C and subsequently supplemented with 400 µM IPTG for 1.5 hr to induce the expression of the BAM complex genes or *dolP*. The envelope fractions were solubilized and purified by Ni-affinity and size exclusion chromatography, adapting a previously published protocol. Briefly, after membrane solubilization with 50 mM Tris-HCl pH 8.0, 150 mM NaCl, and 1%

(w/v) DDM, and removal of insoluble material by ultracentrifugation at 100,000 × $g$, 4°C, soluble proteins were loaded onto a Ni-column (HisTrap FF Crude, GE Healthcare) pre-equilibrated with 50 mM Tris-HCl pH 8.0, 150 mM NaCl, and 0.03% (w/v) DDM (equilibration buffer), using an ÄKTA Purifier 10 (GE Healthcare) at 4°C. The column containing bound proteins was washed with equilibration buffer supplemented with 50 mM imidazole. Proteins were eluted in equilibration buffer, applying a gradient of imidazole from 50 mM to 500 mM and further separated by gel filtration using an HiLoad 16/600 Superdex 200 (GE Healthcare) in equilibration buffer. Eluted proteins were concentrated using an ultrafiltration membrane with a 10 kDa molecular weight cutoff (Vivaspin 6, Sartorius). To reconstitute the BAM–DolP complex in vitro, equimolar concentrations of purified BAM and DolP were used. Purified proteins were mixed in equilibration buffer for 1 hr at 4°C or for 30 min at 25°C as indicated in figure legends. The reaction was further diluted 1:4 times in ice-cold blue native buffer (20 mM Tris-HCl pH 7.4, 50 mM NaCl, 0.1 mM EDTA, 1% [w/v] digitonin, 10% w/v glycerol) and ice cold blue native loading buffer (5% coomassie brilliant blue G-250, 100 mM Bis-Tris-HCl, pH 7.0, 500 mM 6-aminocaproic acid) prior to loading onto home-made 5–13% blue native polyacrylamide gradient gels. Resolved protein complexes were blotted onto a PVDF membrane and immunolabelled. Where non-relevant gel lanes were removed, a white space was used to separate contiguous parts of the same gel.

## Site-directed photo-crosslinking

Cells harbouring pEVOL-pBpF (*Chin et al., 2002*) and pBamA-DolP[His] or pDolP[His] with single amber codon substitutions in the *dolP* open reading frame were cultured in minimal media until early exponential phase, supplemented with 1 mM Bpa (Bachem) and 400 μM IPTG for 1.5 hr. Cultures were divided into two equal parts, one left on ice and one subjected to UV irradiation for 10 min on ice, using a UV-A LED light source (Tritan 365 MHB, Spectroline). Harvested cells were mechanically disrupted to obtain the envelope fraction as described above. Envelope fractions were solubilized in 200 mM Tris-HCl, pH 8, 12% (w/v) glycerol, 4% (w/v) SDS, 15 mM EDTA, and 2 mM PMSF. After a clarifying spin, the supernatants were diluted 20-fold in RIPA buffer (50 mM Tris/HCl, pH 8, 150 mM NaCl, 1% [v/v] NP-40, 0.5%[w/v] sodium deoxycholate, 0.1% [w/v] SDS) supplemented with 20 mM imidazole and subjected to Ni-affinity chromatography. After extensive washing with RIPA buffer containing 50 mM imidazole, proteins were eluted with the same buffer containing 500 mM imidazole. Equal portions of the elution fractions were separated by SDS-PAGE and subjected to immunoblotting.

## Antibodies and western blotting

Proteins separated by SDS-PAGE or blue native-PAGE were transferred onto PVDF membranes (Merck). After blocking with skim milk, membranes were immunolabelled using epitope-specific rabbit polyclonal antisera, with the exception of RpoB that was labelled using a mouse monoclonal antibody (NeoClone Biotechnology). The $F_1\beta$ subunit of the ATP $F_1F_O$ synthase was detected using a rabbit polyclonal antiserum raised against an epitope of the homologous protein of *Saccharomyces cerevisiae* (Atp2). The secondary immunodecoration was conducted using anti-rabbit or anti-mouse antibodies conjugated to horseradish peroxidase produced in goat (Sigma). Protein signals were generated using a Clarity Western ECL blotting substrate (Bio-Rad) and detected using a LAS-4000 (Fujifilm) system. The signal intensities of protein bands were quantified using a Multi Gauge (Fujifilm) software.

## Mass spectrometry analyses

MALDI-TOF MS: Coomassie-stained bands of interest were excised from SDS-polyacrylamide gels and cut into pieces. Samples were washed twice with 100 μl of 25 mM ammonium bicarbonate, 50% (v/v) acetonitrile for 10 min under agitation.

After drying, the gel pieces were rehydrated with 10 μl of 10 μg/ml modified trypsin (Promega) in 25 mM ammonium bicarbonate and digested overnight at 37°C. Acetonitrile was added to the digest to a final concentration of 10% (v/v). After 5 min sonication, 1 μl of the extracted peptide mixture was spotted on the sample plate of the mass spectrometer with 1 μl of the matrix solution (6 mg/ml of α-cyano-4-hydroxycynnamic acid in 50% [v/v] acetonitrile and 0.1% [v/v] trifluoroacetic acid). The analysis was performed using a MALDI-TOF/TOF mass spectrometer (Voyager 5800,

Applied Biosystems/MDS, Sciex) in positive reflectron mode with the following parameters: accelerating voltage, 20 kV; grid voltage, 68%; extraction delay time, 200 ns; shoot number, 1000. Acquisition range was between 750 and 3000 m/z. Spectra were treated using the Data Explorer software (Applied Biosystems).

NanoLC-MS/MS: 70 µg of eluted proteins from each sample (+ UV or – UV) were digested with trypsin (Promega) using S-Trap Micro spin columns (Protifi) according to the manufacturer's instruction (*HaileMariam et al., 2018*). Digested peptide extracts were analysed by online nanoLC using an UltiMate 3000 RSLCnano LC system (ThermoScientific) coupled with an Orbitrap Fusion Tribrid mass spectrometer (Thermo Scientific) operating in positive mode. Five microliters of each sample (5 µg) were loaded onto a 300 µm ID ×5 mm PepMap C18 pre-column (Thermo Scientific) at 20 µl/min in 2% (v/v) acetonitrile, 0.05% (v/v) trifluoroacetic acid. After 3 min of desalting, peptides were online separated on a 75 µm ID × 50 cm C18 column (in-house packed with Reprosil C18-AQ Pur 3 µm resin, Dr. Maisch; Proxeon Biosystems) equilibrated in 90% buffer A (0.2% [v/v] formic acid), with a gradient of 10–30% buffer B (80% [v/v] acetonitrile, 0.2% [v/v] formic acid) for 100 min, then 30–45% for 20 min at 300 nl/min. The instrument was operated in data-dependent acquisition mode using a top-speed approach (cycle time of 3 s). Survey scans MS were acquired in the Orbitrap over 375–1800 *m/z* with a resolution of 120,000 (at 200 *m/z*), an automatic gain control (AGC) target of 4e5, and a maximum injection time (IT) of 50 ms. Most intense ions (2+ to 7+) were selected at 1.6 *m/z* with quadrupole and fragmented by Higher Energy Collisional Dissociation (HCD). The monoisotopic precursor selection was turned on, the intensity threshold for fragmentation was set to 25,000, and the normalized collision energy (NCE) was set to 30%. The resulting fragments were analysed in the Orbitrap with a resolution of 30,000 (at 200 *m/z*), an AGC target of 1e5, and a maximum IT of 100 ms. Dynamic exclusion was used within 30 s with a 10 ppm tolerance. The ion at 445.120025 m/z was used as lock mass. The dipeptides were searched manually in Xcalibur using ms2 reporter ions of the modified peptide (m/z 159.11; 187.11; 244.13) and MSMS spectra of the crosslinked peptides were annotated manually using GPMAW (*Peri et al., 2001*).

## Epifluorescence microscopy and analysis

Overnight cultures of *E. coli* BW25113 and its derivative strains were diluted into fresh M9 medium containing 0.2% glycerol or LB medium and grown at 30°C to $OD_{600}$ = 0.2–0.3. When indicated, cultures were supplemented with 400 µM of IPTG to induce ectopic expression of plasmid-borne genes for 1 hr prior to collecting samples for microscopy analysis. Culture volumes of 0.6 µl were deposited directly onto slides coated with 1% (w/v) agarose in a phosphate-buffered saline solution and visualized by epifluorescence microscopy. Cells were imaged at 30°C using an Eclipse TI-E/B Nikon wide field epifluorescence inverted microscope with a phase contrast objective (Plan APO LBDA 100X oil NA1.4 Phase) and a Semrock filter mCherry (Ex: 562BP24; DM: 593; Em: 641BP75) or FITC (Ex: 482BP35; DM: 506; Em: 536BP40). Images were acquired using a CDD OrcaR2 (Hamamatsu) camera with illumination at 100% from a HG Intensilight source and with an exposure time of 1–3 s, or using a Neo 5.5 sCMOS (Andor) camera with illumination at 60% from a LED SPECTRA X source (Lumencor) with an exposure time of 2 s. Nis-Elements AR software (Nikon) was used for image capture. Image analysis was conducted using the Fiji and ImageJ software. The fraction of cells with $DolP^{GPF}$ signals at mid-cell sites was estimated using the Fiji Cell Counter plugin. Collective profiles of fluorescence distribution versus the relative position along the cell axis were generated using the Coli-Inspector macro run in ImageJ within the plugin ObjectJ (*Vischer et al., 2015*), selecting only cells with a constriction (80% of cell diameter) as qualified objects. Fluorescence intensities were normalized to the mid-cell intensity measured for a control reference strain harbouring the chromosomal *dolP-gfp* fusion.

## Acknowledgements

We thank Harris Bernstein (NIDDK/NIH, Bethesda, MD) for providing key reagents and for comments on the manuscript. We thank Tanneke den Blaauwen (University of Amsterdam) and Nathalie Dautin (IBPC/CNRS, Paris) for discussion. We thank the Light Imaging Toulouse CBI (LITC) platform for assistance with and maintenance of the microscopy instrumentation. We thank Odile Burlet-Schiltz (IPBS/CNRS, Toulouse) for support and access to the Toulouse Proteomics Infrastructure. Further financial support was provided by: the Fondation pour la Recherche Médicale to DR; the Chinese Scholarship

Council as part of a joint international PhD program with Toulouse University Paul Sabatier to YY; the French Ministry of Higher Education and Research to LO-T; the Ecole Normale Supérieure to FR; the European Research Council (Europe Union's Horizon 2020 research and innovation program, grant agreement No 677823), the French governmental Investissement d'Avenir program and the Laboratoire d'Excellence 'Integrative Biology of Emerging Infectious Diseases' (ANR-10-LABX-62-IBEID) to DB; the Centre National de la Recherche Scientifique and the ATIP-Avenir program to RI.

## Additional information

### Funding

| Funder | Grant reference number | Author |
| --- | --- | --- |
| Centre National de la Recherche Scientifique | ATIP-Avenir | Raffaele Ieva |
| Agence Nationale de la Recherche | ANR-10-LABX-62-IBEID | David Bikard |
| Fondation pour la Recherche Médicale | PostDoc Fellowship | David Ranava |
| European Research Council | 677823 | David Bikard |
| École Normale Supérieure | | François Rousset |
| Chinese Scholarship Council | | Yiying Yang |
| Ministère de l'Enseignement Supérieur et de la Recherche | | Luis Orenday-Tapia |

The funders had no role in study design, data collection and interpretation, or the decision to submit the work for publication.

### Author contributions

David Ranava, Conceptualization, Supervision, Validation, Investigation, Visualization, Methodology, Writing - review and editing; Yiying Yang, Luis Orenday-Tapia, Conceptualization, Validation, Investigation, Visualization, Methodology, Writing - review and editing; François Rousset, Software, Formal analysis, Validation, Investigation, Writing - review and editing; Catherine Turlan, Methodology, Validation, Investigation, Writing - review and editing; Violette Morales, Supervision, Validation, Investigation, Methodology, Writing - review and editing; Lun Cui, Cyril Moulin, Carine Froment, Gladys Munoz, Investigation; Jérôme Rech, Investigation, Methodology; Julien Marcoux, Supervision, Validation, Methodology, Writing - review and editing; Anne Caumont-Sarcos, Methodology, Supervision, Validation, Investigation, Visualization, Writing - review and editing; Cécile Albenne, Conceptualization, Supervision, Investigation, Visualization, Writing - review and editing; David Bikard, Conceptualization, Software, Formal analysis, Supervision, Funding acquisition, Methodology, Writing - review and editing; Raffaele Ieva, Conceptualization, Supervision, Funding acquisition, Validation, Investigation, Methodology, Writing - original draft, Project administration, Writing - review and editing

### Author ORCIDs

David Ranava ⬤ http://orcid.org/0000-0002-5841-7699
Luis Orenday-Tapia ⬤ https://orcid.org/0000-0002-1134-0823
Julien Marcoux ⬤ https://orcid.org/0000-0001-7321-7436
David Bikard ⬤ http://orcid.org/0000-0002-5729-1211
Raffaele Ieva ⬤ https://orcid.org/0000-0002-3405-0650

### Decision letter and Author response

Decision letter https://doi.org/10.7554/eLife.67817.sa1
Author response https://doi.org/10.7554/eLife.67817.sa2

## Additional files

### Supplementary files
- Supplementary file 1. List of strains used in this study.
- Supplementary file 2. List of plasmids used in this study.
- Transparent reporting form

### Data availability
All data generated and analysed during this study are available in the manuscript and supporting files. Source data related to the CRISPRi screen are provided in the supporting files.

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
