## [Decision Letter]

**Acceptance summary:**

The refolding of outer membrane proteins is an important component of the bacterial stress response. In this work, the authors demonstrate that DolP interacts with other well studied components of the membrane stress response and is important for proper folding of BamA, an important mediator of protein folding. The manuscript further describes how geospatial relocation of DolP is associated with these newly described functions. Collectively, this study provides novel insight on bacterial outer membrane homeostasis.

**Decision letter after peer review:**

[Editors’ note: the authors submitted for reconsideration following the decision after peer review. What follows is the decision letter after the first round of review.]

Thank you for submitting your work entitled "Outer membrane lipoprotein DolP interacts with the BAM complex and promotes fitness during envelope stress response" for consideration by *eLife*. Your article has been reviewed by 3 peer reviewers, and the evaluation has been overseen by a Reviewing Editor and a Senior Editor. The reviewers have opted to remain anonymous.

Our decision has been reached after consultation between the reviewers. Based on these discussions and the individual reviews below, we regret to inform you that your work will not be considered further for publication in *eLife*.

Reviewers all concur on the high quality and commendable execution of the study, with some interesting findings. However, the associated mechanistic details are not fully resolved with respect to the role of DolP in regulation of membrane biogenesis/homeostasis. The data could also be strengthened with the inclusion of additional controls as pointed out by reviewers.

Reviewer #1:

In this manuscript, Renava et al. explore the function of *E. coli* YraP (renamed DolP by the authors), a poorly understood lipoprotein that is upregulated during the σ^E^ envelope stress response for unknown reasons. Consistent with previous studies showing that DolP is required for envelope integrity, the authors show that a dolP deletion strain is sensitive to vancomycin. They then use an elegant CRISPRi-based synthetic phenotype screen to show that the silencing (or partial silencing) of genes that encode factors that promote cell division, regulate the σ^E^ stress response, or promote efficient assembly of outer membrane proteins (OMPs) exacerbate defects associated with the deletion of dolP. In the second half of the paper, they show that although DolP does not play a direct role in OMP assembly, it becomes important when OMP assembly is impaired or when BamA, a major subunit of the OMP assembly machinery (BAM) that binds to DolP, is overexpressed and thereby titrates DolP away from the septum.

Although the experiments presented in this study were nicely executed and the data are of a high quality, I had trouble fitting all of the experimental observations into a convincing model that explains the function of DolP. The authors propose that DolP is a cell division factor that also prevents detrimental effects of excessive BAM production during cell stress. This is a nice idea that is supported by the data, but it does not explain why the deletion of dolP causes an outer membrane defect that increases sensitivity to vancomycin in unstressed cells. My main concern is that although the results show that DolP interacts with BamA and describe a potentially important function for DolP under very specific conditions, they otherwise provide limited insight into the general function of DolP and its mechanism of action.

Additional comments:

1. Based on structural data, the BAM lipoproteins bind to the POTRA domains of BamA that are close to the membrane and likely hinder the access of other proteins, so it is not entirely clear how a small lipoprotein like DolP would interact with BamA. Does DolP prevent the binding of the lipoproteins or force their dissociation? Further information on the site of interaction would help to address this question and perhaps provide further insight into DolP function.

2. Some of the genes that are silenced in the CRISPRi screen are essential or nearly essential for cell viability (e.g., BamD, FtsEX), so presumably they are only partially silenced. The authors should clarify this point and perhaps add some data that indicates how effectively the levels of expression are reduced.

Reviewer #2:

The manuscript relates to the regulation of bacterial outer membrane remodelling and biogenesis. This is important because the integrity of the envelope is essential for survival, yet it needs to be remodelled during bacterial cell division. If this process is not well managed then the bacteria become vulnerable. Thus, the regulation of outer membrane biogenesis and the remodelling of the protein, peptidoglycan and lipid components during cell division is critical for survival and growth.

Specifically, the work focusses on role of DolP (renaming of YraP) in this regulatory process, including the related response that is associated with defective outer membrane biogenesis – such as the build-up of unfolded outer membrane protein.

The paper is very interesting and highlights some very interesting interactions – most notably the interaction with DolP and the BAM machinery of the outer membrane. The experiments have been superbly executed and the results are clean and clear. However, the conclusions are somewhat vague. The documented properties of DolP, while believable and interesting, do not lead to a clear mechanistic understanding of its regulation of outer membrane biogenesis. Likewise the connection between the behaviour of DolP and the cellular response to envelope stress, impaired outer-membrane protein biogenesis and cell division are interesting, but there is no new understanding of its underlying mechanism.

Could the authors deliver more detail or further clarification along the lines of one or more of the following:

How does DolP affect the activity of Bam? And what are the consequences of this? How does that affect outer membrane biogenesis? And how is this chain of events modulated during envelope stress and cell division?

The authors point out a very interesting interaction between DolP and the Bam machinery. But the direct consequences of this interaction are not entirely clear.

This reviewer would be very interested to know how the interactions and properties of DolP affect outer membrane biogenesis and exactly what role it plays in the response to envelope stress and cell division.

Reviewer #3:

The manuscript is clearly written and interesting. The experiments are well done, with the appropriate controls. Overall, the work is very solid and the data convincing. However, although the authors report interesting observations and raise intriguing hypothesis on the function of DolP, they fail to identify it and provide little mechanistic insight.

The interaction between DolP and the Bam machinery is mostly shown using immunoblots, which are very sensitive. The authors used periplasmic and inner membrane proteins as controls (also OmpA in one of the experiment). However, they need to include additional outer membrane (lipo)proteins as controls.

The depletion of DolP from mid-cell under stress is interesting; the authors need to show that this correlates with an increase in the levels of the DolP-BamA complex (using cross-linking for instance).

The authors should comment more on the link with the amidases.

The authors should normalise the levels of BamA and BamA∆P1 used in the experiment shown in Figure 4C (difficult to compare lanes 2-3 to 6-7)

The plots shown in Figure 5 B and C are convincing. They should also be shown in Figure 5A.

---

## [Author Response]

[Editors’ note: the authors resubmitted a revised version of the paper for consideration. What follows is the authors’ response to the first round of review.]

Reviewers all concur on the high quality and commendable execution of the study, with some interesting findings. However, the associated mechanistic details are not fully resolved with respect to the role of DolP in regulation of membrane biogenesis/homeostasis. The data could also be strengthened with the inclusion of additional controls as pointed out by reviewers.Reviewer #1:In this manuscript, Renava et al. explore the function of *E. coli* YraP (renamed DolP by the authors), a poorly understood lipoprotein that is upregulated during the σE envelope stress response for unknown reasons. Consistent with previous studies showing that DolP is required for envelope integrity, the authors show that a dolP deletion strain is sensitive to vancomycin. They then use an elegant CRISPRi-based synthetic phenotype screen to show that the silencing (or partial silencing) of genes that encode factors that promote cell division, regulate the σE stress response, or promote efficient assembly of outer membrane proteins (OMPs) exacerbate defects associated with the deletion of dolP. In the second half of the paper, they show that although DolP does not play a direct role in OMP assembly, it becomes important when OMP assembly is impaired or when BamA, a major subunit of the OMP assembly machinery (BAM) that binds to DolP, is overexpressed and thereby titrates DolP away from the septum.Although the experiments presented in this study were nicely executed and the data are of a high quality, I had trouble fitting all of the experimental observations into a convincing model that explains the function of DolP. The authors propose that DolP is a cell division factor that also prevents detrimental effects of excessive BAM production during cell stress. This is a nice idea that is supported by the data, but it does not explain why the deletion of dolP causes an outer membrane defect that increases sensitivity to vancomycin in unstressed cells. My main concern is that although the results show that DolP interacts with BamA and describe a potentially important function for DolP under very specific conditions, they otherwise provide limited insight into the general function of DolP and its mechanism of action.

We thank this reviewer for the positive comments on the quality of our work. The explanation as to why DolP is important for OM integrity is likely to be multifactorial. It was proposed that the OM integrity defect of Δ*dolP* cells could be caused, at least in part, by the impaired NlpD-mediated activation of AmiC (Tsang et al., Plos Genetics 13:e1006888, 2017). In our revised manuscript, we reveal that DolP supports BamA folding, providing additional insight into why DolP is critical form OM integrity.

In response to comments of all three reviewers, we have further addressed the mechanism by which DolP promotes envelope integrity. To this end, we have carried out new sets of experiments:

– We have shown that, in a strain that undergoes envelope stress such as Δ*bamB,* lack of DolP causes a reduction of the levels of BamA, inferring that DolP is critical for BamA stability, at least under stress conditions (new Figure 2D).

– We had already revealed in our first submission that the overaccumulation of the BAM complex in the OM is particularly detrimental in cells lacking DolP. This experimental setup was exploited to assay the function of DolP under an envelope stress condition. In the revised version of our manuscript we extend this analysis. We show that a more marginal accumulation of BamA, which does not cause toxicity, enhances the OM permeability to vancomycin (new Figure 3E). We also demonstrate that OM integrity is restored by the concomitant increase of the levels of DolP expression (new Figure 3E). Overall, the fact that DolP opposes the detrimental effects of BamA overproduction (new Figures 3D and 3E) suggests that DolP is critical for the correct functioning of BamA.

– We have carried out additional experiments to obtain insight into the molecular bases of how DolP influences BamA. First we have investigated why the accumulation of BamA causes a detrimental effect. Fractionation of cells that overproduce BamA and heat-modifiability analysis of their protein content have revealed that overproduced BamA efficiently accumulates in the OM, however a large portion of this protein is improperly folded (new Figure 3F and Figure 3-figure suppl. 1D). To understand how DolP opposes the BamA detrimental effect, we have conducted the same experiment with cells that overproduce both BamA and DolP. Notably, we found that the increased expression of DolP restores proper folding, and to some extent functioning, of BamA (new Figures 3F and 3G).

– As discussed in another response to a point of the reviewers, we have used site-directed photo-crosslinking to prove the proximity of DolP to BamA in vivo (new Figure 4C).

Taken together our work demonstrates that DolP interacts with BamA and supports its proper folding in the OM. These observations provide a plausible explanation as to why DolP is critical for OM integrity and to cope with envelope stress.

Additional comments:1. Based on structural data, the BAM lipoproteins bind to the POTRA domains of BamA that are close to the membrane and likely hinder the access of other proteins, so it is not entirely clear how a small lipoprotein like DolP would interact with BamA. Does DolP prevent the binding of the lipoproteins or force their dissociation?

We agree with the reviewer that the organization of DolP with the BAM subunits is a very interesting point to investigate. We have addressed the interaction of DolP with the BAM complex using different methodologies, in vivo and in vitro. Our native pull-down analysis reveals that at least BamA, BamD, BamC and BamE can be co-isolated with DolP (Figure 4A). We could not address whether BamB was also co-isolated with DolP as we do not have an antiserum that specifically recognizes this subunit of the BAM complex. However, to investigate if DolP can interact with the BAM holo-complex (including all its 5 core subunits), we have reconstituted the interaction in vitro and we have analyzed the product of this reaction by blue native gel electrophoresis. We have compared the molecular mass of the purified BAM complex alone to that of the BAM complex supplemented with DolP (Figures 4D and 4E). Our immunoblots with either DolP or BamA antibodies have revealed a complex that migrates slower than BAM in samples supplemented with DolP. Because DolP (20 kDa) is significantly smaller than BamB (42 kDa), the slower migration of the BAM-DolP complex compared to BAM alone strongly suggests that DolP interacts with the BAM holo-complex. In the revised manuscript, we have slightly modified the description of this result to better highlight this important nuance in the interpretation of our blue native analysis of the BAM-DolP complex.

Finally, it should be noted that previous biochemical and structural studies have reported *i*) that both the β-barrel and the periplasmic domains of BamA are highly dynamic; *ii*) that the BAM lipoproteins interact with BamA mainly via interactions with POTRA3 in proximity of the linker with POTRA2 (in the case of BamB), and via POTRA 5 (in the case of BamCDE). In particular, a domain-dissection analysis of the BamA periplasmic POTRA motifs indicated that POTRA5, but not the other POTRA motifs, is a key requirement for the association of BamCDE to BamA (Kim et al., Science 317:961-4, 2007). Together, these observations suggest that several subdomains of BamA may be available for an interaction with a relatively small protein, such as DolP (20 kDa), without necessarily causing the dissociation of the BAM lipoproteins. Thus, in our opinion, it does not appear surprising that the association of DolP with BamA does not abolish its interaction with the other lipoproteins of the complex.

Further information on the site of interaction would help to address this question and perhaps provide further insight into DolP function.

We very much agree with the reviewer that mapping of the DolP-BamA interaction can be potentially insightful. To address this point, we have performed in vivo site-directed photo-crosslinking, showing that DolP positions V52, V72 and, to a lower extent, V101, are proximal to BamA (new Figure 4C and Figure 4-figure suppl. 1B). Although we have used mass-spectrometry in the attempt to identify the amino acid residues of BamA crosslinked by DolP, this approach was not successful. Nevertheless, we have obtained information on the site of the DolP three-dimensional structure that makes contact to BamA. In fact, positions V52, V72 and V101 are proximal in the three-dimensional structure of DolP with their side chains oriented away from the BON2 domain and exposed on the surface of the BON1 domain (new Figure 4C). Thus, these results identify in BON1 a site of interaction with BamA. Whereas the C-terminal BON2 domain of DolP was shown to bind phospholipids (Bryant et al., *eLife* 9, 2020), the role of the BON1 domain was unclear. Our result provides an additional piece of information on the organization of DolP at the BAM complex suggesting that BON1 can be important for the association of DolP with BamA.

2. Some of the genes that are silenced in the CRISPRi screen are essential or nearly essential for cell viability (e.g., BamD, FtsEX), so presumably they are only partially silenced. The authors should clarify this point and perhaps add some data that indicates how effectively the levels of expression are reduced.

The efficiency of gene silencing by dCas9 depends on the guide RNAs that targets specific genetic features. To address the point raised by the reviewer, we have conducted a western blot showing the silencing effect by 2 guide RNAs specific for BamD (essential gene) and BamE (non-essential gene), respectively. Upon expressing dCas9 for 17 generations, (as conducted in the genome-wide screen presented in Figures 1B and 1C), the expression of both *bamD* and *bamE* was reduced (new Figure 1-figure suppl. 2D), although the silencing effect was more prominent for *bamE*. A residual production of BamD explains why repression of *bamD* was not lethal in our drop tests. However, this degree of *bamD* repression was sufficient to cause a synthetic growth defect when combined with the deletion of *dolP* (Figure 1D).

Reviewer #2:The manuscript relates to the regulation of bacterial outer membrane remodelling and biogenesis. This is important because the integrity of the envelope is essential for survival, yet it needs to be remodelled during bacterial cell division. If this process is not well managed then the bacteria become vulnerable. Thus, the regulation of outer membrane biogenesis and the remodelling of the protein, peptidoglycan and lipid components during cell division is critical for survival and growth.Specifically, the work focusses on role of DolP (renaming of YraP) in this regulatory process, including the related response that is associated with defective outer membrane biogenesis – such as the build-up of unfolded outer membrane protein.The paper is very interesting and highlights some very interesting interactions – most notably the interaction with DolP and the BAM machinery of the outer membrane. The experiments have been superbly executed and the results are clean and clear. However, the conclusions are somewhat vague. The documented properties of DolP, while believable and interesting, do not lead to a clear mechanistic understanding of its regulation of outer membrane biogenesis. Likewise the connection between the behaviour of DolP and the cellular response to envelope stress, impaired outer-membrane protein biogenesis and cell division are interesting, but there is no new understanding of its underlying mechanism.

We thank this reviewer for the comments about the quality of our experimental approaches and the relevance of our observations. We have taken into consideration the constructive criticism of the reviewer and we have further characterized the mechanisms by which DolP interacts with BamA and contributes to maintain OM integrity. The new mechanistic insight into the function of DolP that we have obtained has helped to strengthen our conclusions, and overall our revised manuscript is better focused on the role of DolP in OM homeostasis.

Could the authors deliver more detail or further clarification along the lines of one or more of the following:How does DolP affect the activity of Bam? And what are the consequences of this? How does that affect outer membrane biogenesis?

In response to comments of all three reviewers, we have further addressed the mechanism by which DolP promotes envelope integrity. To this end, we have carried out new sets of experiments:

– We have shown that, in a strain that undergoes envelope stress such as Δ*bamB,* lack of DolP causes a reduction of the levels of BamA, inferring that DolP is critical for BamA stability, at least under stress conditions (new Figure 2D).

– We had already revealed in our first submission that the overaccumulation of the BAM complex in the OM is particularly detrimental in cells lacking DolP. This experimental setup was exploited to assay the function of DolP under an envelope stress condition. In the revised version of our manuscript we extend this analysis. We show that a more marginal accumulation of BamA, which does not cause toxicity, enhances the OM permeability to vancomycin (new Figure 3E). We also demonstrate that OM integrity is restored by the concomitant increase of the levels of DolP expression (new Figure 3E). Overall, the fact that DolP opposes the detrimental effects of BamA overproduction (new Figures 3D and 3E) suggests that DolP is critical for the correct functioning of BamA.

– We have carried out additional experiments to obtain insight into the molecular bases of how DolP influences BamA. First we have investigated why the accumulation of BamA causes a detrimental effect. Fractionation of cells that overproduce BamA and heat-modifiability analysis of their protein content have revealed that overproduced BamA efficiently accumulates in the OM, however a large portion of this protein is improperly folded (new Figure 3F and Figure 3-figure suppl. 1D). To understand how DolP opposes the BamA detrimental effect, we have conducted the same experiment with cells that overproduce both BamA and DolP. Notably, we found that the increased expression of DolP restores proper folding, and to some extent functioning, of BamA (new Figures 3F and 3G).

– As discussed in another response to a point of the reviewers, we have used site-directed photo-crosslinking to prove the proximity of DolP to BamA in vivo (new Figure 4C).

Taken together our work demonstrates that DolP interacts with BamA and supports its proper folding in the OM. These observations provide a plausible explanation as to why DolP is critical for OM integrity and to cope with envelope stress.

And how is this chain of events modulated during envelope stress and cell division?The authors point out a very interesting interaction between DolP and the Bam machinery. But the direct consequences of this interaction are not entirely clear.

As stated in response to the previous comment, in our revised manuscript we provide evidence that DolP interacts with BamA and supports its proper folding and functioning.

To better understand what modulates the DolP-BamA interaction, first we have used an in vivo site-directed photo-crosslinking approach to identify contact sites between DolP and BamA. Subsequently, we have mimicked a feature of the OM of cells undergoing envelope stress and monitored the DolP-BamA interaction.

– First, DolP positions V52, V72 and, to a lower extent, V101, were found to be crosslinked to BamA. Notably, positions V52, V72 and V101 are proximal in the three-dimensional structure of DolP with their side chains oriented away from BON2 and exposed on the surface of BON1 (new Figure 4C). These results identify in BON1 a site of interaction of DolP with BamA.

– Second, we found that the same DolP site that interacts with BamA can also interact with OmpA (new Figure 4C and Figure 4-figure suppl. 2A-C). The mass spectrometry analysis identified in the C-terminal periplasmic domain of OmpA the site of interaction of DolP (new Figure 4-figure suppl. 2D-F). Importantly, by mimicking a feature of the envelope stress response (depletion of OmpA), we have obtained evidence that OmpA competes with BamA for an interaction with DolP (new Figure 5D). Thus, during envelope stress the depletion of OmpA might play an important role by favouring the association of DolP with BamA. The notion that OmpA plays a role in buffering the function of a stress response factor was recently suggested in another study. OmpA was proposed to modulate the signalling function of a periplasmic stress response factor (RcsF) via a competition mechanism (Dekoninck et al., *eLife* 9, 2020). In light of those and our results, we like to believe that OmpA may play a more general role in buffering the function of envelope stress response factors that would become readily available as soon as the levels of OmpA are reduced as a consequence of stress.

This reviewer would be very interested to know how the interactions and properties of DolP affect outer membrane biogenesis and exactly what role it plays in the response to envelope stress and cell division.

In the revised manuscript, we shown that cells that accumulate improperly folded BamA in their membrane fraction present an OM integrity defect which is rescued by the overproduction of DolP (Figure 3E). We have obtained some mechanistic insights into this rescuing function of DolP. First, the accumulation of improperly folded BamA causes a reduction of the levels of OmpA and OmpC (new Figure 3G). Second, overproduced DolP restores proper folding of BamA and to some extent the normal levels of OmpA and OmpC (new Figures 3E and 3G). These results suggest that, by improving the folding of BamA in the OM, DolP indirectly promotes efficient OMP biogenesis, thereby restoring OM integrity. We believe these results are revelatory of a mechanism by which DolP supports the BAM complex in meeting the high demand for OMP assembly under a stress condition.

Finally, we have demonstrated that envelope stress as well as the overaccumulation of BamA impairs the recruitment of DolP at cell constriction sites during a late step of cell division, and that this effect is specific for DolP. Reduced levels of DolP at cell constriction sites have a negative effect on the regulation of septal peptidoglycan hydrolysis (Tsang et al., Plos Genetics 13:e1006888, 2017). Further studies will be warranted to better understand how DolP influences septal peptidoglycan hydrolysis. In our work, however, we have focused mainly on the role of DolP during envelope stress and in particular we have addressed the function of the DolP-BamA interaction. In addition to the new results that we have incorporated in the revised manuscript, we have modified the Abstract, the final paragraph of the Introduction and the Discussion to highlight our new mechanistic findings and to better focus our revised manuscript on the role of DolP during envelope stress.

Reviewer #3:The manuscript is clearly written and interesting. The experiments are well done, with the appropriate controls. Overall, the work is very solid and the data convincing. However, although the authors report interesting observations and raise intriguing hypothesis on the function of DolP, they fail to identify it and provide little mechanistic insight.

We appreciate the positive comments of the reviewer on the quality of our work and on the clarity of our manuscript. We have taken into consideration the criticism of this reviewer and we have conducted new experiments to address the function of DolP.

In response to comments of all three reviewers, we have further addressed the mechanism by which DolP promotes envelope integrity. To this end, we have carried out new sets of experiments:

– We have shown that, in a strain that undergoes envelope stress such as Δ*bamB,* lack of DolP causes a reduction of the levels of BamA, inferring that DolP is critical for BamA stability, at least under stress conditions (new Figure 2D).

– We had already revealed in our first submission that the overaccumulation of the BAM complex in the OM is particularly detrimental in cells lacking DolP. This experimental setup was exploited to assay the function of DolP under an envelope stress condition. In the revised version of our manuscript we extend this analysis. We show that a more marginal accumulation of BamA, which does not cause toxicity, enhances the OM permeability to vancomycin (new Figure 3E). We also demonstrate that OM integrity is restored by the concomitant increase of the levels of DolP expression (new Figure 3E). Overall, the fact that DolP opposes the detrimental effects of BamA overproduction (new Figures 3D and 3E) suggests that DolP is critical for the correct functioning of BamA.

– We have carried out additional experiments to obtain insight into the molecular bases of how DolP influences BamA. First we have investigated why the accumulation of BamA causes a detrimental effect. Fractionation of cells that overproduce BamA and heat-modifiability analysis of their protein content have revealed that overproduced BamA efficiently accumulates in the OM, however a large portion of this protein is improperly folded (new Figure 3F and Figure 3-figure suppl. 1D). To understand how DolP opposes the BamA detrimental effect, we have conducted the same experiment with cells that overproduce both BamA and DolP. Notably, we found that the increased expression of DolP restores proper folding, and to some extent functioning, of BamA (new Figures 3F and 3G).

– As discussed in another response to a point of the reviewers, we have used site-directed photo-crosslinking to prove the proximity of DolP to BamA in vivo (new Figure 4C).

Taken together our work demonstrates that DolP interacts with BamA and supports its proper folding in the OM. These observations provide a plausible explanation as to why DolP is critical for OM integrity and to cope with envelope stress.

We have simplified Figure 2-figure suppl. 1, the scope of which is to show that DolP does not play a direct role in OMP assembly. As we show in the subsequent part of the manuscript, this role is linked to proper BamA folding and functioning, which might be particularly important to meet the high demand for OMP assembly under a stress condition.

The interaction between DolP and the Bam machinery is mostly shown using immunoblots, which are very sensitive. The authors used periplasmic and inner membrane proteins as controls (also OmpA in one of the experiment). However, they need to include additional outer membrane (lipo)proteins as controls.

To address this point of the reviewer, we have included additional controls for the immunoblots of our pull-downs (OmpA immunoblot in Figure 4A and LamB immunoblot in Figure 4-figure suppl. 1A). Most importantly, the concern of the reviewer about the specificity of the DolP-BamA interaction has been addressed by proving the vicinity of DolP to BamA with an in vivo site-directed photo-crosslinking method (new Figure 4C).

The depletion of DolP from mid-cell under stress is interesting; the authors need to show that this correlates with an increase in the levels of the DolP-BamA complex (using cross-linking for instance).

We very much appreciate this comment that we have addressed with new experiments. First we have used in vivo site-directed photo-crosslinking and we have identified in the BON1 domain of DolP a site of interaction with BamA. We found also that the same site of DolP interacts with the C-terminal periplasmic domain of OmpA (new Figure 4C, Figure 4-figure suppl. 1B, Figure 4-figure suppl. 2). Our crosslink experiment is conducted in a strain deleted of chromosomal *dolP* and harbouring plasmid-borne *dolP*^amber^ (conditionally expressed when Bpa is supplemented to the growth medium). Given that lack of DolP causes a strong fitness defects under envelope stress conditions, performing the same crosslinking experiment upon inducing stress, as suggested by the reviewer, would pose a number of important experimental challenges. Instead, we have conducted our crosslink experiment in a strain that mimics a feature of the envelope stress response, and in particular the depletion of OmpA. Our results indicate that OmpA competes with BamA for the interaction with DolP (new Figure 5D). In cells lacking OmpA, we have also noticed a slight reduction of the DolP^GFP^ signal at the mid-cell as well as a small increment of the signal at non-septal position. Under envelope stress conditions, the drastic reduction of the levels of OmpA might favour the association of DolP with BamA.

As stated in response to reviewer 2, the notion that OmpA plays a role in buffering the function of a stress response factor was recently suggested in another paper. OmpA was proposed to modulate the signalling function of the periplasmic stress response factor (RcsF) via a competition mechanism (Dekoninck et al., *eLife* 9, 2020). In light of those and our results, we like to believe that OmpA may play a more general role in buffering the function of a periplasmic stress-response factors that would become readily available as soon as the levels of OmpA are reduced as a consequence of stress.

The authors should comment more on the link with the amidases.

We agree with the reviewer that the link with amidases is very interesting. We demonstrate that envelope stress, as well as the overaccumulation of BamA, impair the recruitment of DolP at cell constriction sites during a late step of cell division, and that this effect is specific for DolP. Presumably, reduced levels of DolP at cell constriction sites may have a negative effect on the regulation of septal peptidoglycan hydrolysis. Further studies are warranted to better understand the role of DolP in the regulation of septal peptidoglycan hydrolysis. Our revised work, however, focuses mainly on the role of DolP during envelope stress and in particular addresses the function of the DolP-BamA interaction. We have modified the Abstract, the final paragraph of the Introduction and the Discussion to highlight our new mechanistic findings and to better focus our manuscript on the role of DolP during envelope stress.

The authors should normalise the levels of BamA and BamA∆P1 used in the experiment shown in Figure 4C (difficult to compare lanes 2-3 to 6-7)

As suggested by the reviewer, we have quantified these results, normalizing the amount of BamA in the elution to the amount of BamA in the total fractions. Our quantifications clearly confirm our initial conclusion that the deletion of POTRA1 reduces the amount of BamA that is assembled into the OM. The new data are shown in the legend of the Figure 4B.

The plots shown in Figure 5 B and C are convincing. They should also be shown in Figure 5A.

The fluorescence collective profile plot of cells overproducing all the BAM subunits has been added to Figure 5A. Similarly to BamA, the overproduction of all BAM subunits impairs the localization of DolP at cell constriction sites, thus confirming our initial conclusions.